# ACTIVE PROBABILISTIC REASONING IN HUMANS AND LANGUAGE MODELS

## ABSTRACT

Can large language models (LLMs), when acting as agents, match human cognitive capabilities in sequential reasoning? To answer this question, we designed a novel active probabilistic reasoning task that can be played by humans and LLMs. Our minimal task design allows us to disentangle two essential components of decision-making, sampling (gathering evidence) and inference (evaluating evidence). We evaluated a large set of LLMs and find a wide spectrum of performance. Several frontier models reach human-level performance, but do not exceed skilled human players. Strong model performance consistently relies on extensive reasoning. While some LLMs outperform humans in inference, all models consistently lag in sampling capabilities. To probe the source of these differences, we develop a novel Bayesian modeling framework that tracks sampling-policy updates and maps humans and LLMs to different classical observer models. We show that humans tend toward maximum-a-posteriori (MAP) sampling, whereas the best LLMs tend to minimize posterior entropy across options. We further tested whether LLMs can improve via in-context learning, and found that only a subset of top-performing models could learn to solve the task based only on the outcome of their choices.

## 1 INTRODUCTION

Neuroscience and cognitive science have long intersected in the effort to explain cognition by linking neural mechanisms to mental processes (Sloman et al., 2021; van Bree, 2024). A key step in establishing this link is the development of quantitative models of cognitive processes, which provide insights into the computations underlying cognition and reveal latent variables that are more directly related to neural processes than overt behavior (Ji-An et al., 2025; Miller et al., 2023; Yang and Wang, 2020; Richards et al., 2019). Recent advances in Artificial Intelligence (AI) have lead to entirely new avenues to develop such models (Lake et al., 2017; Peterson et al., 2021). Large Language Models (LLMs), in particular, have been trained at unprecedented scales of data and compute. These models show human-level performance across a variety of tasks (Bommasani, 2021; Bubeck et al., 2023), exhibit emergent properties such as in-context learning (ICL) (Brown et al., 2020; Olsson et al., 2022), can be prompted to generate explicit reasoning traces (Wei et al., 2022; Kojima et al., 2022), and are subject to alignment techniques designed to steer the model toward human-like behavior (Bai et al., 2022; Griffith et al., 2013; Wei et al., 2021; Ouyang et al., 2022). Fine-tuning of LLMs on experimental human datasets Binz et al. (2025) can produce models that not only replicate human behavioral statistics but also approximate classical neuro-scientific models of decision-making. As a result, researchers have begun to propose LLMs as candidate models of cognition (Binz and Schulz, 2023). While these advances highlight the potential of LLMs to serve as mechanistic models of cognition, in most settings it remains unclear how LLMs solve a particular task, and how their strategies compare to those employed by humans. Most current evaluations of LLMs focus on complex reasoning benchmarks, such as mathematics (Glazer et al., 2024), logic and problem solving (Rein et al., 2024; Wang et al., 2024; Phan et al., 2025; Yue et al., 2024; White et al., 2025), or code generation (Jimenez et al., 2024; Yang et al., 2025a), where performance is typically assessed by final-answer accuracy, while leaving the underlying mechanisms and dynamics largely unexplored.

To address this gap, here we study the behavior of humans and LLMs in a novel task probing their abilities in sequential probabilistic reasoning. Our task disentangles two core processes underlying many complex, goal-oriented behaviors in humans and animals: the active gathering

of evidence from the environment (sampling) and the integration of potentially unreliable evidence towards an understanding of the unknown rules governing the environment (inference). A large body of research suggests that cognition broadly arises from generalizations of these two processes, and that neural computations may be understood as adaptations optimized to support them (Friston, 2012; Kepecs and Mainen, 2012; Gershman, 2018; Knill and Pouget, 2004). The minimal environment of our task enables Bayesian modeling of optimal policies, allowing precise quantification of sampling and inference capabilities across human and LLM agents. This framework reveals both commonalities and divergences in their strategies, moving beyond end-point accuracy to provide a deeper evaluation of LLMs as potential models of cognition.

**Related work:** *Human-LLM comparisons.* Using LLMs as cognitive models (Binz and Schulz, 2023). Comparative work links LLM behaviour and learning to human psychophysics tasks (Russin et al., 2025; Binz et al., 2025) and the ability of models fine-tuned to human behavior to reach Bayesian like behavior on $k$-armed bandit tasks (Su et al., 2025). *In-context mechanisms.* Transformers can implement algorithmic updates during the forward pass, matching gradient descent or ridge regression on linear tasks (Akyürek et al., 2022; von Oswald et al., 2022), and can be trained to in-context learn broad function classes (Garg et al., 2022), including preconditioned behavior (Ahn et al., 2023; Fu et al., 2024). Martingale tests report deviations from Bayesian scaling (Falck et al., 2024), while other probes and prompting regimes can induce approximately Bayesian choices (Gupta et al., 2025) and ICL as a Bayesian process Xie et al. (2022).

**Contributions (i)** We introduce an *active probabilistic reasoning task* that disentangles *sampling* (evidence acquisition) from *inference* (evidence integration), enabling direct human–LLM comparison and interpretable, model-based analysis. **(ii)** We evaluate a broad set of contemporary LLMs, spanning different architectures, sizes and training paradigms, against human participants under identical instructions, revealing a graded performance spectrum: several LLMs reach human-level performance but do not surpass the best human participants. **(iii)** We quantify *sampling* and *inference quality*, showing that LLMs can exceed humans in inference capabilities, yet consistently under-perform in *sampling*. **(iv)** We characterize how agents integrate evidence by tracking the posterior probability of final agent choice across rounds, showing that human-like integration in LLMs exists and relies on extended chain-of-thought reasoning. **(v)** We test LLMs' in-context learning capabilities with different prompt variations and find heterogeneous outcomes: while some models show brittleness over long horizons, others exhibit clear gains, with improvements largely attributable to extended reasoning effort. **(vi)** We develop and fit a unifying observer class with an interpolating geometric policy update that recovers natural and Euclidean policy gradients. Fitted policies map humans near MAP-like sampling, whereas top LLMs trend toward entropy-modulated strategies.

## 2 ACTIVE PROBABILISTIC REASONING TASK

We introduce an active probabilistic reasoning task (Fig. 1) drawing inspiration from classical psychophysics and k-armed bandit paradigms (Lai and Robbins, 1985; Daw et al., 2006; Najemnik and Geisler, 2005), that explicitly separates sampling from inference. Each trial (i.e., independent game) consists of a random number of sampling rounds ($N \in 2, \ldots, 15$) followed by a single integration round. In each sampling round, the agent selects one of 4 buttons, with a randomly sampled number of occlusions between 0 and 3. A chosen button reveals a cue from one of two classes: RED and GREEN. At the start of each trial, one of the 4 buttons is designated as the *biased* one, yielding a RED cue with probability 0.9 and GREEN cue with probability 0.1, while the remaining buttons produce cues uniformly (0.5/0.5). We further extend this pipeline to different tasks, enabling the evaluation of language models across a variety of stochastic environments (see Appendix F). To implement this task, we developed an *online platform*[1] through which human participants can play, while LLMs interact with an equivalent *text-based* version under identical instructions (see Fig. 1). Notably, the platform provides a *leaderboard*, allowing human players to compare their performance both with other participants and with LLMs.

---

[1] https://ai.trt-bench.org

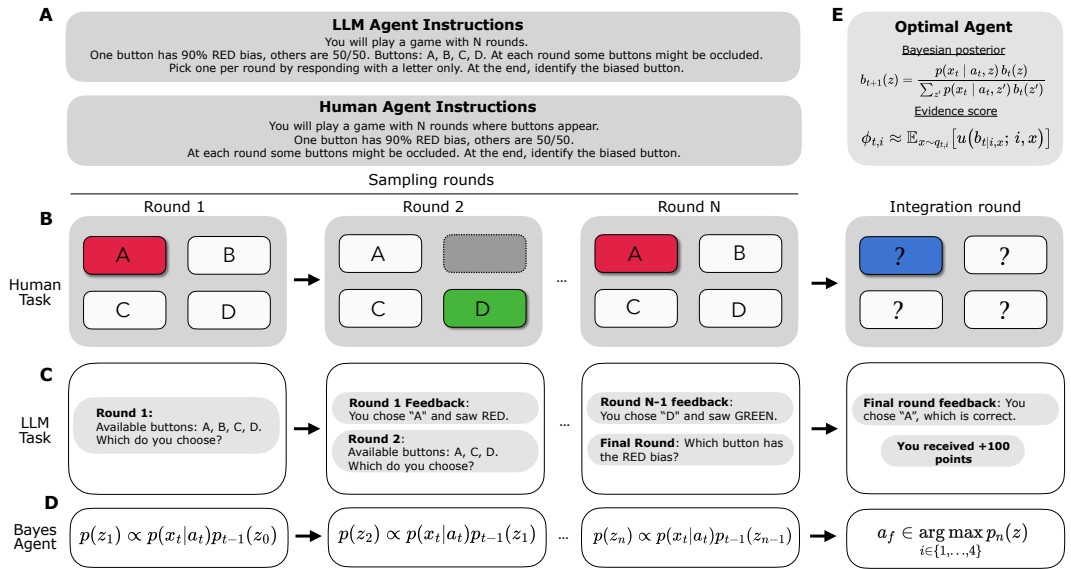

Figure 1: **Structure of active probabilistic reasoning task**. **A**: Instructions provided to human and LLM agents. Humans and LLM receive an analogous explanatory prompt before starting the game. **B**: Schematic of the online platform template for humans. The participants sample from a set of 4 buttons: A, B, C or D (with occlusions in dark grey). Upon click, a RED- or GREEN-colored cue is revealed. After $N$ rounds, the participant is asked which button has the highest bias towards the RED cue. **C**: Analogous schematic for the *text-based* version of the task for LLMs. **D and E**: We compare agents choices during both the sampling and inference rounds to various ideal observers with a Bayesian belief updated at each round. Each observer is defined by a different scoring function $\phi$ which integrates a Bayesian belief $b_t$ over the sampled evidence.

## 3 COMPARING HUMAN AND SYNTHETIC COGNITION

We evaluate a broad set of Large Language Models (LLMs) on the proposed task (Figure 2). The models span most of the current LLM landscape, ranging from *dense* to *Mixture-of-Experts* (MoE) architectures across a range of model sizes and training paradigms, from *base* models to *instruct*-fine-tuned, *reasoning*, and *hybrid-reasoning* LLMs (Vaswani et al., 2017; Schulman et al., 2017; Shoeybi et al., 2019; Wei et al., 2021; Ouyang et al., 2022; Wei et al., 2022; Shu et al., 2023; Shao et al., 2024; Cai et al., 2025). Our assessment covers both state-of-the-art closed-source systems and competitive open-weight models. We evaluate a wide range of well-known model families. This includes OpenAI's *gpt 4o mini* (Hurst et al., 2024), *gpt 4.1 mini* (OpenAI, 2025a), *gpt 5 mini* (OpenAI, 2025b), and the *gpt oss* open architectures in both the 20B and 120B parameter variants (OpenAI, 2025c). We further considered several *llama* models (Touvron et al., 2023; Dubey et al., 2024), including variants fine-tuned on human behavioral data (Binz et al., 2025), as well as a distilled version of *deepseek* (Guo et al., 2025). Our assessment also covered Anthropic's *claude sonnet 4* and *claude haiku 3.5* (Anthropic, 2025), Google's *gemini 2.5 pro/flash* and the smaller *gemma* models (Comanici et al., 2025; Gemma Team, Google DeepMind, 2025), and the *qwen* family, including the 235B Mixture-of-Experts (MoE) model as well as earlier dense variants (Qwen Team, 2025; Yang et al., 2025b). Finally, we included the fully open-source *apertus* model (Hernández-Cano et al., 2025), *grok 3 mini* (xAI, 2025), and *glm 4.5* (Zeng et al., 2025). A subset of the *reasoning models* considered allows for control over *reasoning effort* (resulting in longer or shorter *chain-of-thoughts* token streams), hence we evaluate their performance both in *low* and *high* parameter condition, we report this as additional *Extended Thinking* bars in Fig. 2. For every LLM and *reasoning effort* level, we evaluate a minimum of $1,400$ individuals games spanning uniformly the 2 to 15 rounds range, amounting to more than $55,000$ games. For what concerns humans, we performed data collection on 50 human subjects during a 1-hour live-competition setting amounting to $5000$ individual games played spanning uniformly the same 2 to 15 rounds range. Beyond assessing task performance, this evaluation helps chart new directions in understanding

which architectures and training paradigms may give rise to viable substrates for cognitive modeling.

The average success rate among human participants is $61\%$, matching the performance of the best LLM, *gpt 5 mini*. However, the top $25\%$ of human players outperform this model by a margin of $7\%$. Examining how success rates evolve across rounds (Fig. 2B) reveals a clear separation between models around the $45\%$ mark. These two groups exhibit strikingly different behaviors: one shows positively sloped curves, indicating that the models leverage longer games (# of rounds) to improve their performance, while the other remains flat across rounds. Within the ranking, *claude haiku 3.5* is the first model to display this *lift-off*. From *DeepSeek R1 Qwen3 8B* onward, all models closely follow the success profiles of the lower $75\%$ of human players. Interestingly, this subgroup aligns with the *reasoning* models. To account for the probabilistic nature of the task, we consider a complementary metric assessing the agreement of the agents' choices at *integration rounds* with that of an optimal Bayesian observer (Figure 2C). We build a Bayesian posterior over the evidence sampled by agents (see Appendix B for details), estimate the optimal choice according to a maximum-a-posteriori (MAP) agent, and compute the average agreement between the agents and MAP observer. Concretely, given a Bayesian posterior over a latent variable $z \in \mathbb{R}_+^K$, and a set of actions $a \in 1, ..., K$ with $K = 4$, the MAP agent is defined at round $t$ by a belief $b_t$, defined in this case as the Bayesian posterior over the evidence and a policy $\pi_t^{\mathrm{MAP}}$. We can then calculate the *Bayesian agreement* score at round $T$, $S_T$ (shown in Fig. 2C and 3A) is defined, for a trial with length $T$ as

$$\pi_t^{\mathrm{MAP}} = \arg\max_{i \in \{1, ..., K\}} b_t(z = i) \quad , \; S_T = \frac{1}{N} \sum_{i=1}^{N} \mathbb{I}\left[ \mathrm{a}_T = \pi_T^{MAP} \right] \tag{1}$$

with the indicator function is comparing the match of human/LLM agent and MAP choices.

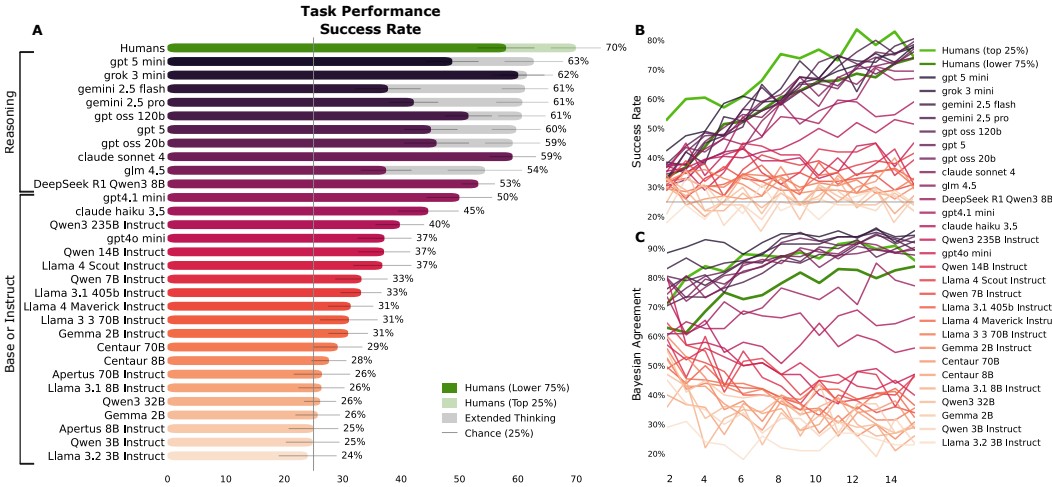

Figure 2: **Agents performance on activate probabilistic reasoning task**. **A**: Task performance based on success rate. Human performance (green) is split into the lower $75\%$ and the top $25\%$ of participants. LLM performance is shown as colored bars, grouped by model type (*base*, *instruct*, and *reasoning*) with *low, absent, or non-controllable reasoning effort*. Models with *high reasoning effort* (*extended thinking*) are shown as grey bars. Error bars represent standard deviations, computed across trial-cluster means with a uniform distribution over the number of rounds. Overall, *reasoning* models outperform *base* and *instruct* variants. Human participants achieved an average success rate of about $61\%$, comparable to the best LLM (*gpt 5 mini*), while the top $25\%$ of humans exceeded it by $7\%$. **B**: Evolution of success rate across rounds. Models are color-coded by their average success rate from panel **A**. Human participants are shown in light green (top $25\%$) and dark green (lower $75\%$). For reasoning models allowing *extended thinking* we report only *high reasoning effort* performance. **C**: Evolution of Bayesian agreement reports the matching of agents' choices to the MAP decision based on the evidence they sampled across rounds.

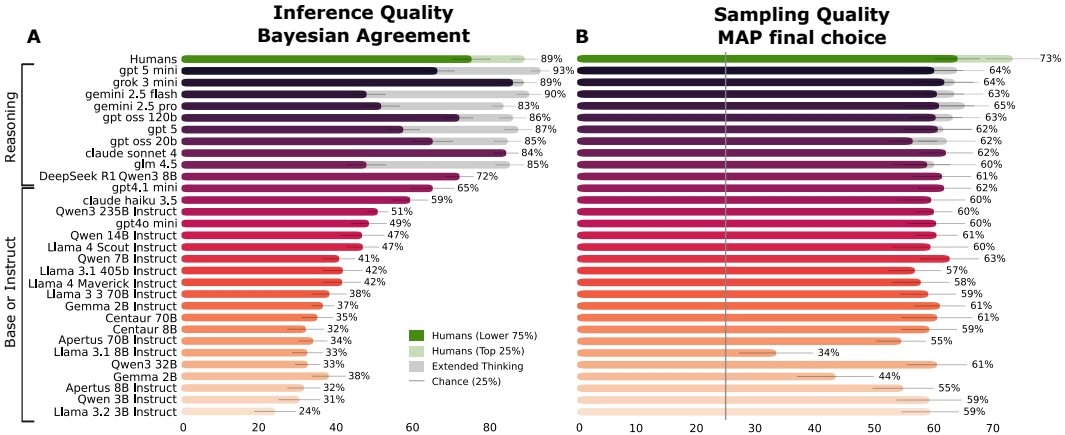

Figure 3: **Agents Inference and Sampling Quality**. **A**: Inference quality of human and LLM agents based on Bayesian agreement. Human performance (green) is split into the lower $75\%$ and the top $25\%$ of participants. LLM performance is shown as colored bars, grouped by model type (*base*, *instruct*, and *reasoning*) with *low, absent, or non-controllable reasoning effort*. Models with *high reasoning effort* (*extended thinking*) are shown as grey bars. Error bars represent standard deviations, computed across trial-cluster means with a uniform distribution over the number of rounds. Bars report the average matching of agents' choices to the MAP decision based on the evidence they sampled across rounds. Top LLMs (*gpt 5 mini*, *grok 3 mini*, *gemini 2.5 flash*) match or surpass the best human players. **B**: Sampling quality of human and LLM agents based on success rate with MAP optimal final choice. We observe modest differences across LLMs and only limited benefits from increased *reasoning effort*. In contrast, the top $25\%$ of human participants consistently outperform all models.

The analysis of Bayesian agreement assesses agents' *inference quality* by measuring their ability to integrate evidence gathered during *sampling rounds* relative to an optimal Bayesian observer. In particular, the evolution of Bayesian agreement, shown in Figure 2C, confirms the existence of distinct performance profiles among LLMs, with top-performing models aligning with the top $25\%$ of human participants. When results are averaged across the number of rounds (Figure 3A), *gpt 5 mini*, *grok 3 mini*, and *gemini 2.5 Flash* match or even surpass the best human players in their agreement with the MAP agent, particularly in longer games. This evaluation underscores the central role of *reasoning effort*, operationalized as the generation of additional *chain-of-thought* tokens, in supporting successful evidence integration.

However, one question remains unanswered: if Bayesian agreement highlights the superiority of reasoning models over base and instruct variants, and even over humans, in integrating evidence, where do humans retain an advantage over these models? To address this, we examine the quality of agents' *sampling strategies*. Concretely, we estimate the Bayesian posterior over the evidence sampled by each agent, derive the MAP-optimal choice, and compute the corresponding average success rate. This rate measures the maximum performance that a given agent could have achieved given the samples it actually collected. As shown in Figure 3B, this analysis reveals a more uniform distribution of success rates, indicating only modest differences in the quality of sampling strategies across LLMs. It further suggests that *reasoning effort* plays only a limited role in this regard, providing slim performance gains, whereas the top $25\%$ of human participants consistently outperform all LLMs.

## 4 LANGUAGE MODELS SHOW HUMAN-LIKE POSTERIOR INTEGRATION

The analyses in Figures 2–3 revealed that top-performing LLMs match or outperform humans in inference quality. To better characterize how agents integrate evidence, we examine the evolution of their Bayesian posteriors over the course of sampling. Figure 4 reports the average posterior dynamics for games of fixed length, with trajectories showing how evidence accumulates toward the model choice in the final integration round. Each curve reflects the probability mass assigned to

the ultimately chosen option as additional samples are observed.

Remarkably, we observe that LLMs can display evidence integration patterns that closely mirror those of humans. A first observation is that model size alone does not predict posterior quality. Large models such as *Qwen3 235B Instruct* fail to form well-structured evidence profiles, trailing behind both humans and smaller models (panel **C**). By contrast, compact models equipped with sufficient reasoning capacity can closely approximate the posterior dynamics of top-performing LLMs and human participants. In successful agents, posterior trajectories separate quasi-monotonically across trial lengths: more rounds lead to more sharply defined posteriors, reflecting systematic integration of accumulating evidence. Poorer models, in contrast, exhibit a form of *mode collapse* (panel **D**): their posteriors remain flat in short trials and only show weak differentiation when the maximum number of rounds is reached.

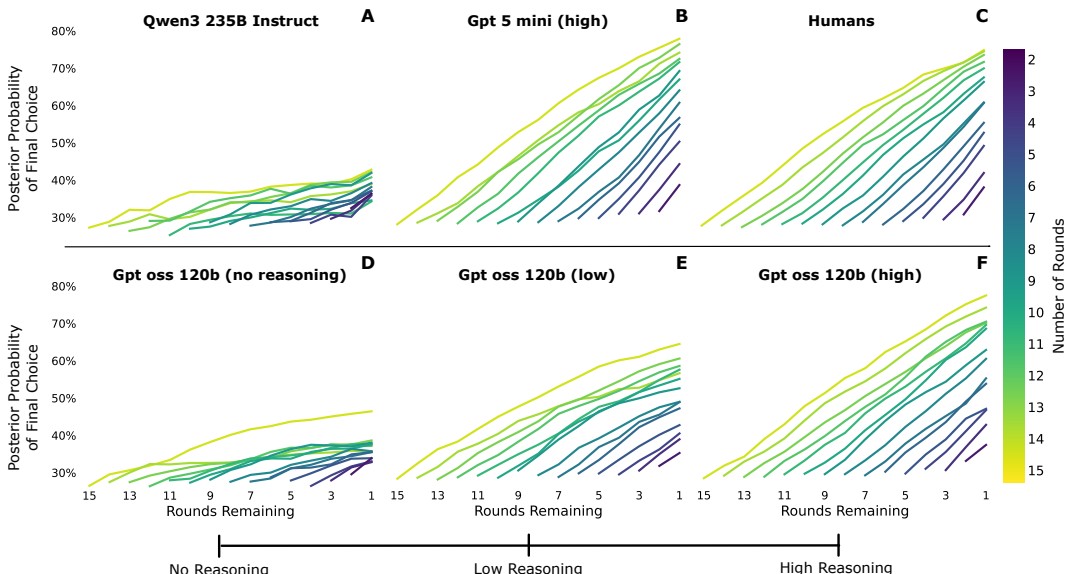

Figure 4: **Posterior evolution across rounds and reasoning levels**. Each curve shows the posterior probability of the final choice as evidence accumulates over rounds. **A**: Large *Qwen3 235B Instruct* model exhibits weak integration. **B**: High-reasoning *gpt 5 mini* shows rapid early growth and high final confidence. **C**: Humans display comparable accelerated integration with late plateau on longer games. **D-F**: Variants of *gpt oss 120b* highlight the effect of *reasoning effort*: with *no reasoning* (**D**) posteriors tend to remain flat; *low reasoning* (**E**) yields moderate gains; high reasoning (**F**) produces sharp separation and human-like trajectories. Overall, stronger reasoning effort shifts posterior gains earlier and raises final confidence, distinguishing reasoning models and humans from baseline LLMs. Figures reporting all model and reasoning levels combinations are included in Appendix Fig. 9, 8.

*Reasoning effort* emerges as a key determinant of posterior quality. For instance, *gpt oss 120b* displays near-random profiles under *low* reasoning, but transitions to human-like evidence integration as reasoning depth increases (*medium* and *high*). A similar effect is seen in the best-performing model, *gpt 5 mini*, where posterior trajectories become sharply separated only when extended reasoning is enabled. This pattern confirms that the task requires non-trivial computation: without sufficient reasoning, models remain on par with non-reasoning baselines, but with increased reasoning effort they converge toward the posterior profiles observed in skilled humans.

## 5 REASONING IMPROVES PERFORMANCE DURING IN-CONTEXT-LEARNING

Thus far, our analyses used a single-trial (comprising $N$ rounds) setup with a prompt (termed here *original prompt*, previously described in Fig. 1) that explicitly states the latent priors of the task (one cue 90/10, others 50/50) and a clear goal (i.e., *which button had the highest ratio of RED?*). This effectively provides models with an optimal prior. To study learning dynamics under minimal task

information, we also designed a *minimal prompt* that only specifies allowed choices (A–D), omitting priors, bias structure, and rewards (see Appendix E for prompt details). This minimal version removes optimal priors and goal framing, requiring models to infer task regularities directly from experience. We therefore evaluate in-context learning (ICL) across both prompting regimes, asking whether models can refine their performance under the *original prompt* and discover latent structure under the *minimal prompt*, as measured over repeated trials (Fig. 5). We run sequences of 1–15 trials, each consisting of 15 rounds, with the transcript of completed trials carried over as context for subsequent ones (Fig. 5A). For the purpose of this analysis, we restrict the model space to include the top-performing closed- and open-source models: *gpt 5 mini, grok 3 mini, gemini 2.5 flash, gpt oss 20b and 120b*. For each model, *reasoning effort* level and *prompt variation* we evaluated at least 100 simulations amounting to more than $30,000$ individual games. We then evaluate performance using the metrics defined in Sections 3, 4: *success rate* (Fig. 5 B-E), *inference quality* (Bayesian agreement) (Fig. 5 C-F), *sampling quality* (Appendix Fig. 10 C-F), and the evolution of posterior trajectories to capture changes in evidence accumulation toward the final choice across in-context trials (Fig. 5 D,G, and Appendix Fig. 11 for all models).

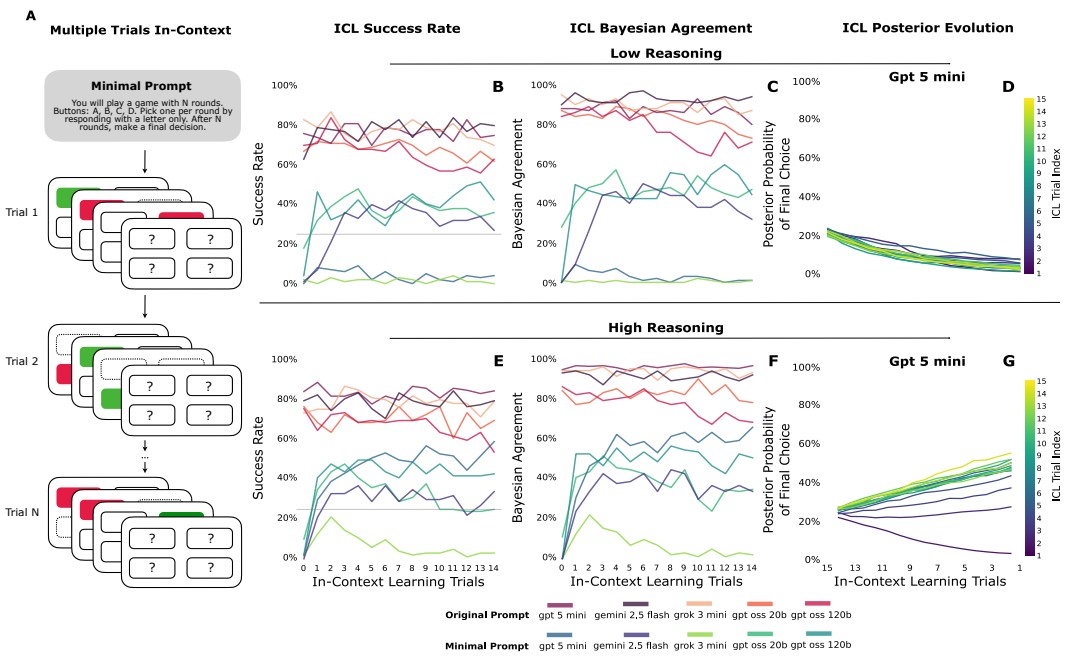

Figure 5: **In-context learning across repeated trials A**: Schematic of multi-trial in-context setup. Each trial transcript is carried over as context to the next. **B, E**: Success under the *original prompt* remains near baseline or degrades at both reasoning levels, while the *minimal prompt* shows heterogeneous outcomes with models capable of ICL and others below chance level, with improved ICL performance for *gpt 5 mini* under high reasoning effort. **D**: Posterior dynamics for *gpt 5 mini* under low reasoning and minimal prompt: probability assigned to the final choice declines across trials, indicating impaired evidence integration. **C, F**: The Bayesian agreement analysis (panels **C**, **F**) corroborates the patterns of success rate. **G**: Posterior dynamics for *gpt 5 mini* under high reasoning and minimal prompt: posterior mass on the final choice strengthens across trials, reflecting improved evidence accumulation.

Under the *original prompt*, success rates exhibit little systematic improvement across in-context trials, remaining close to the baseline single trial averages (Fig. 5 B–E). Notably, the open-source *gpt oss 20b* and *120b* variants show a progressive decline after approximately six trials (panel **B**), this degradation is delayed in the *20B* model and attenuated in the *120B* model under increased reasoning effort (panel **E**). The Bayesian agreement analysis (panels **C**, **F**) corroborates this pattern, demonstrating that extended reasoning stabilizes belief updating and yields consistently high-alignment with the Bayesian observer. Under the *minimal prompt*, both *grok 3 mini* and *gpt 5 mini* initially fail

to perform the task, falling below chance level in terms of both success rate and Bayesian agreement (panels **B**, **C**). By contrast, other models display clear signs of ICL, gradually improving their performance across trials. For *gemini 2.5 flash* and the *gpt oss* variants, performance remains largely stable across reasoning levels (panels **E**, **F**), while *grok 3 mini* shows only weak gains and remains below chance-level. However, increasing reasoning effort markedly enhances *gpt 5 mini*, which rises to top-tier performance on both success rate and Bayesian agreement, with strong evidence of ICL. The posterior dynamics (panels **D**, **G**) highlight this contrast: whereas low reasoning led to impaired integration, extended reasoning enables evidence accumulation that improves across trials under the minimal prompt.

# 6 HUMANS AND LANGUAGE MODELS SHOW DIFFERENT SAMPLING STRATEGIES

Our earlier analyses showed that, while LLMs can integrate evidence as well as or better than humans, their sampling strategies are weaker. To better understand this difference, we compare human and model behavior by fitting a set of observer models with distinct sampling policies both to human and LLM data. We anchor each *observer* with the same belief $b_t$ defined in Sec. 3, a policy on the simplex $\Delta \in \mathbb{R}$, $\pi_t \in \Delta^{K-1}$, and score function $\phi_t \in \mathbb{R}_+^K$, where $K = 4$ is the number of buttons of the task. The policy in this observer model is implemented through logits $\theta_t \in \mathbb{R}^K$

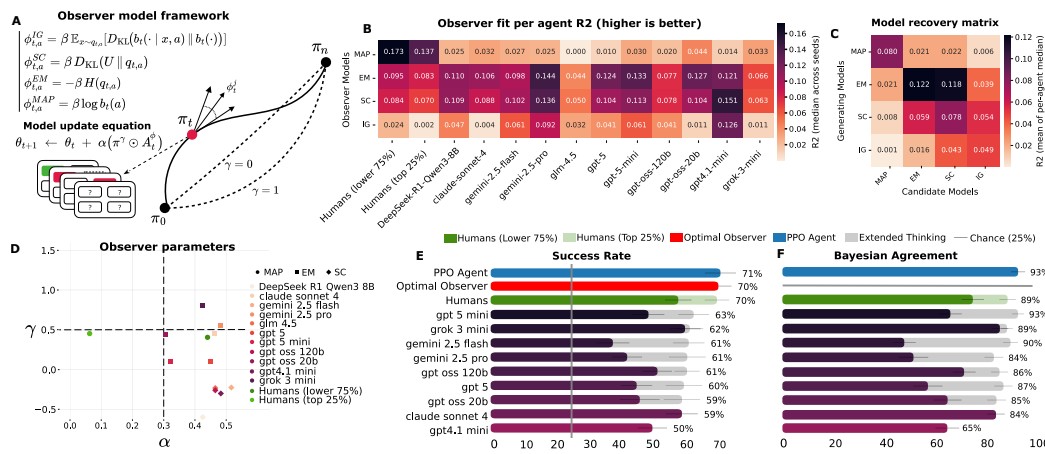

Figure 6: **Human participants and LLMs show different sampling policies A:** Schematic of the gradient interpolating model that was fitted to both humans and language models, different $\gamma$ lead to different curvatures of the gradient trajectories over the simplex. **B:** $R^2$ of the fit across the different observer classes *(MAP, EM, SC, IG)* for each of the top performing models. Humans show a larger preference for MAP-like sampling policies and language models for Entropy modulated policies. **C:** Model recovery matrix for the fitted dataset showing the recovery of the initial fit from generated data of the observers with the highest $R^2$. **D:** Parameter span for $\alpha$ (learning rate) and $\gamma$ (gradient interpolating factor) for the top performing models. Top $25\%$ humans show as outliers in the parameter space. Models show higher variability in the interpolating factor than in the learning rate parameter. **E-F:** Evaluation of the trained PPO agent and the best performing Optimal Observer (MAP) on the same task. Both models show human-like performance. The PPO model fit to the task validates the performance range of our theoretical observer model for the success rate on the task.

by $\pi_t = \text{softmax}(\theta_t)$ and through the bayesian posterior beliefs $b_t$. In order to simulate different decision making strategies, the model is able to differentially reweigh the evidence and change its sampling policy directly on the logits as they go through this gradient descent. This gradient is defined by an *advantage* $A_t^\phi \in \mathbb{R}_+^K$ which at each timestep computes the distance of the *observer score* $\phi$, which evaluates current and future evidence to the goal $J(\pi) = \sum_{i=1}^K \langle \pi, \phi_i \rangle$. Concretely this advantage is given by $A_t^\phi := \phi_t - \langle \pi_t, \phi_t \rangle \mathbf{1}$ with $\langle \pi, \phi \rangle = \sum_{i=1}^K \pi_i \phi_i$. Additionally, in order to explore the types of gradient updates being performed by the agents we unify the Euclidean and Fisher geometries by endowing the logit space with an interpolating metric $G_\gamma(\theta_t) = \text{diag}(\pi_t^{1-\gamma})$,

which leads to the following gradient update equation for the logits $\theta$ (proof in Appendix B.7)

$$\theta_{t+1} = \theta_t + \alpha\left(\pi_t^\gamma \odot A_t^\phi\right). \tag{2}$$

This allows us to fit different gradient profiles within the statistical manifold of the underlying simplex of this task. It can be directly seen that this general expression reduces to the natural-gradient update for $\gamma = 0$, $\Delta\theta = \alpha A_t^\phi$ (Amari, 2016; Kakade, 2001; Kerimkulov et al., 2025), and to the Euclidean policy-gradient update for $\gamma = 1$, $\Delta\theta = \alpha(\pi_t \odot A_t^\phi)$ (Sutton et al., 2000). Classical observers can be integrated into this model by particular choices of the score $\phi_t$. Different canonical observers can then be defined by their corresponding score function $\phi$. We define a MAP agent, (Lai and Robbins, 1985), an *Entropy-Minimizing* agent EM by $\phi^{EM}$, a *Self-Certainty* agent SC by $\phi^{SC}$(Zhao et al., 2025) and a *Information Gain* agent with score function defined by $\phi_{t,a}^{IG}$ (Gershman, 2019). As these different scoring functions execute different computations over the evidence built by the posterior, the gradient will also evolve differently through the perturbations of the advantage $A_t^\phi$. In order to get an idea of the potential explanatory power of this model, we did an initial fit to the active probabilistic reasoning task of all the *observer model* types and compared it also to an on-policy MLP network with PPO(Schulman et al., 2017), showing in panels **E, F**[2]. We then fit this model to the games played by the human players and a subset of the top performing LLMs in order to disentangle their sampling strategies. We show the $R^2$ score over random seeds of the trials dataset for each agent. Notably, we see a sharp distinction between human players and LLMs, with humans having the best fit for MAP-like sampling policies (see model fitting details in Appendix C). LLMs match more closely sampling strategies that are sensitive to the evolution of the posterior entropy, preferring overall EM and SC-like strategies. Furthermore, their gradient parameterization shows an interesting pattern; the top human players seem to adapt with smaller learning rates than the remaining bottom 75% of humans and language models. Overall the gradients executed lie in the middle between a fully natural gradient and a policy gradient as can be seen in Figure 6D.

## 7 CONCLUSION

In Section 2, we introduced an *active probabilistic reasoning task* enabling direct comparison between humans and LLMs. In Section 3, we showed that while some models reach or exceed human-level in inference capabilities, their sampling strategies remain consistently weaker. We observe striking similarities between LLM and human agents in their posterior integration trajectories where reasoning through *chain-of-thought* plays a critical role (Section 4). Further, Section 6 shows that human and model behaviors map to distinct observer classes, with humans preferentially adopting MAP-like strategies, whereas LLMs rely more on entropy-driven sampling policies. These findings raise interesting questions as to how these differences emerge: whether they are related to model architectural choices and what training paradigms can be used to steer model behaviors toward human-like sampling strategies. The minimal environment of our task, and the possibility to derive optimal Bayesian policies, make of it a suitable framework to answer these questions.

Our in-context-learning experiments in Section 5 demonstrate the ability of these models to create a basic representation of the environment, exemplified by their performance improvements, even under settings with minimal information. Notably, the in-context learning experiments reveal a further dichotomy between the abilities for sampling and inference across different models: whereas inference quality improves in-context for some models, the sampling quality remains largely fixed throughout the in-context learning window. While these findings point to only a limited ability for in-context learning, even the partial improvements we observed support the idea that LLMs could become suitable candidates for simulators of cognition on decision-making tasks, as proposed in (Binz et al., 2024).

Beyond leading to precise insights into the abstract computational algorithms implemented by LLMs during *active probabilistic reasoning*, we believe that our task holds great promise as a tool towards *mechanistic interpretability* of LLMs (Elhage et al., 2021; 2022; Olsson et al., 2022). Our analysis reveals key latent variables accounting for a large fraction of the models' sampling choices and inference, which are both based on properties of the Bayesian posterior probabilities. In analogy to neuroscience studies, the identification of correlates of these latent variables in model activity can provide a starting point for understanding the underlying mechanisms.

---

[2]Bayesian agreement not shown for the optimal observer model as it shares the same mechanism as the underlying Bayesian scoring metric.

## 8 REPRODUCIBILITY STATEMENT

To ensure reproducibility of our results, all code used in our experiments, along with detailed instructions for setup and execution, is available at: `https://drive.google.com/drive/folders/17tQxO02lLN1VpwbOF_IIiM9oSm8DmRik`. Additionally, the active probabilistic reasoning task used for data collection is accessible at `https://ai.trt-bench.org`.

## 9 ETHICS STATEMENT

We obtained institutional ethics approval for the collection, analysis, and publication of data collected as part of this study. All participants provided informed consent after receiving study information. Data were anonymized prior to analysis and access was restricted to the research team. The no-risk cognitive task posed no safety concerns to participants.

## 10 LANGUAGE MODEL USE STATEMENT

We used large language models to polish prose and to surface related scientific work; all study design, analysis, and conclusions were generated by the authors, and models were not used to generate data, or results; queries contained only non-sensitive text, outputs were edited for accuracy and style, and all citations were independently verified.

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

# A  ADDITIONAL FIGURES

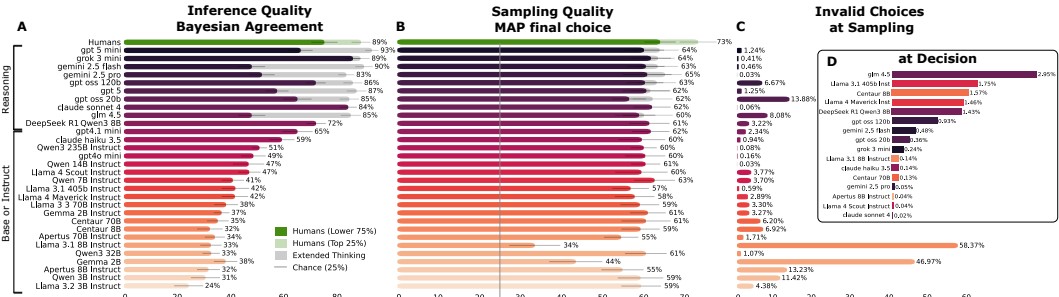

Figure 7: **Extended benchmark analysis of inference, sampling, and invalid choices. A-B**: inference quality (Bayesian agreement with the MAP observer) and sampling quality (success conditional on MAP-optimal choice). Human performance (green) is split into the lower 75% and the top 25% of participants; LLM performance is shown as colored bars, grouped by model type. Models with high reasoning effort (extended thinking) are shown as gray bars. **C**: the proportion of invalid choices, measured both during sampling rounds and at the final integration step. Invalid responses do not correlate strongly with overall success, indicating they are not the primary driver of model performance differences. High performing models like *gpt oss 20b/120b* have a high invalid choice sampling rate yet retain a high overall success rate.

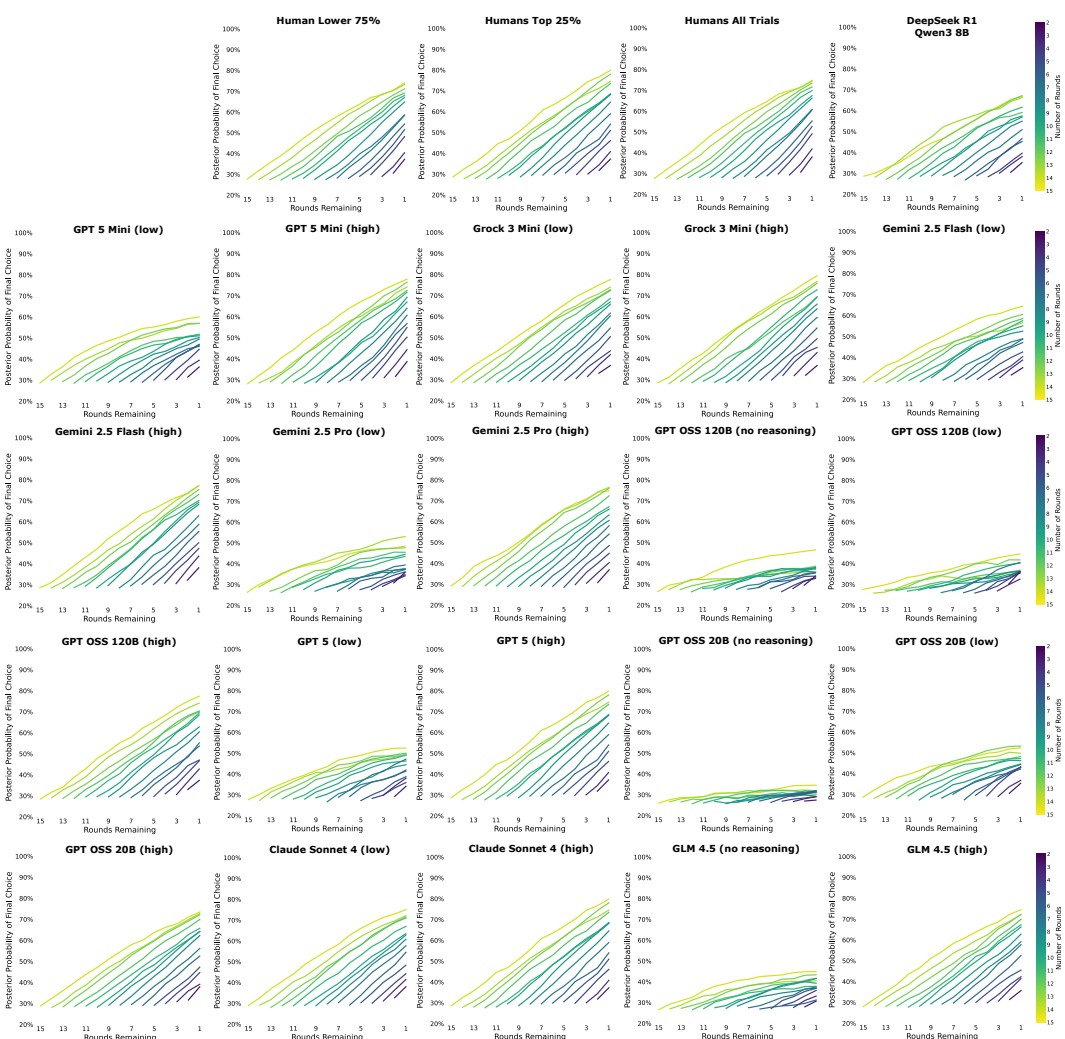

Figure 8: **Posterior evolution by rounds across humans and reasoning models.** Each curve shows the posterior probability assigned to the final choice as additional evidence is sampled, averaged across trials of the same length. Humans exhibit steady evidence accumulation, with the top 25% of participants showing clearer separation and stronger late-round convergence than the lower 75%. Overall, increasing *reasoning effort* shifts gains earlier in the trajectory and raises final confidence, indicating faster and more stable integration.

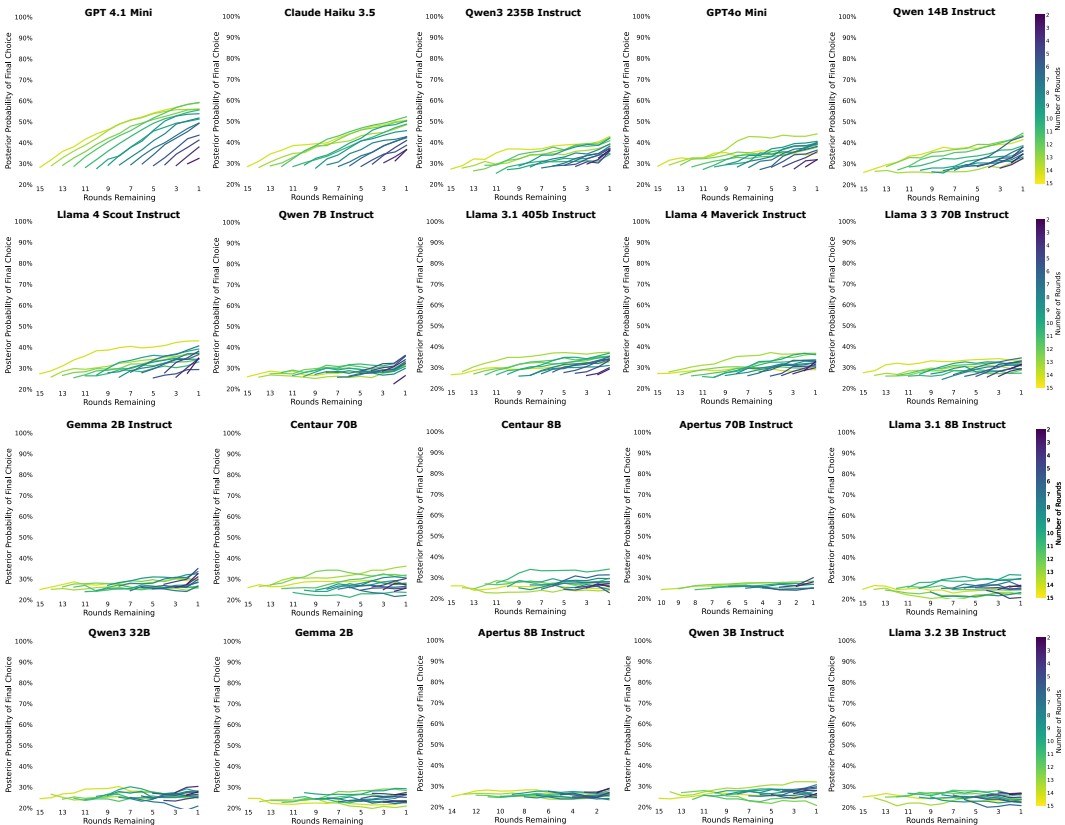

Figure 9: **Posterior evolution by rounds for base and instruct models.** Results are provided in task success rate rank order (left → right, row-wise ↓). Models are ordered by overall task success (left→right, row-wise). Relative to reasoning models (Figure 4; Figure 8), base/instruct variants show shallow growth and delayed or weak separation across trial lengths, indicating poor integration of sampled evidence even when longer rounds are available. As success rate decreases, posteriors increasingly resemble random baselines, with little growth in confidence even after many rounds.

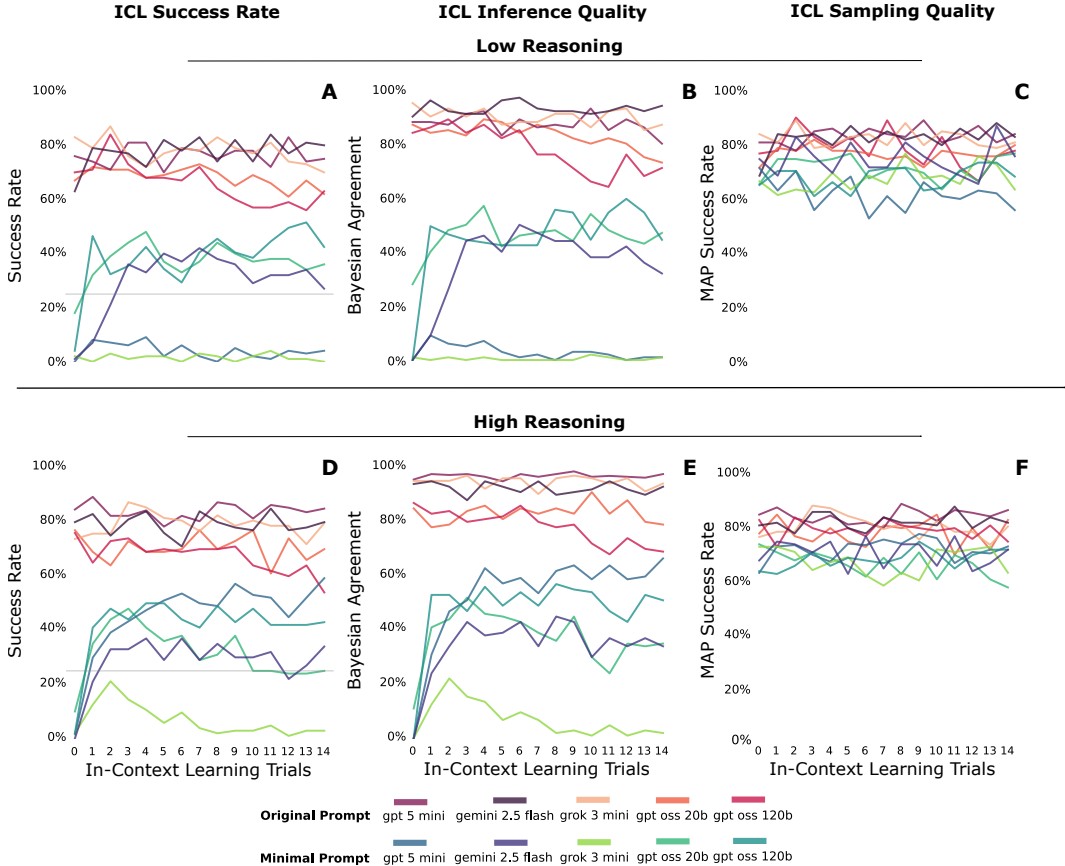

Figure 10: **In-context learning success rate, inference quality, and sampling quality by reasoning levels.** Each panel reports performance across repeated trials under the *original* and *minimal* prompt conditions. **A-B**: in the low reasoning condition, top models show limited or no improvement over trials, with substantial prompt dependent variability. **D-E**: by contrast, high reasoning stabilizes both inference and sampling dynamics, yielding steady gains across trials and reduced spread between prompts. **C, F** shows that in-context improvements and prompt variability arise primarily from enhanced *inference quality* rather than fundamental changes in *sampling strategy*.

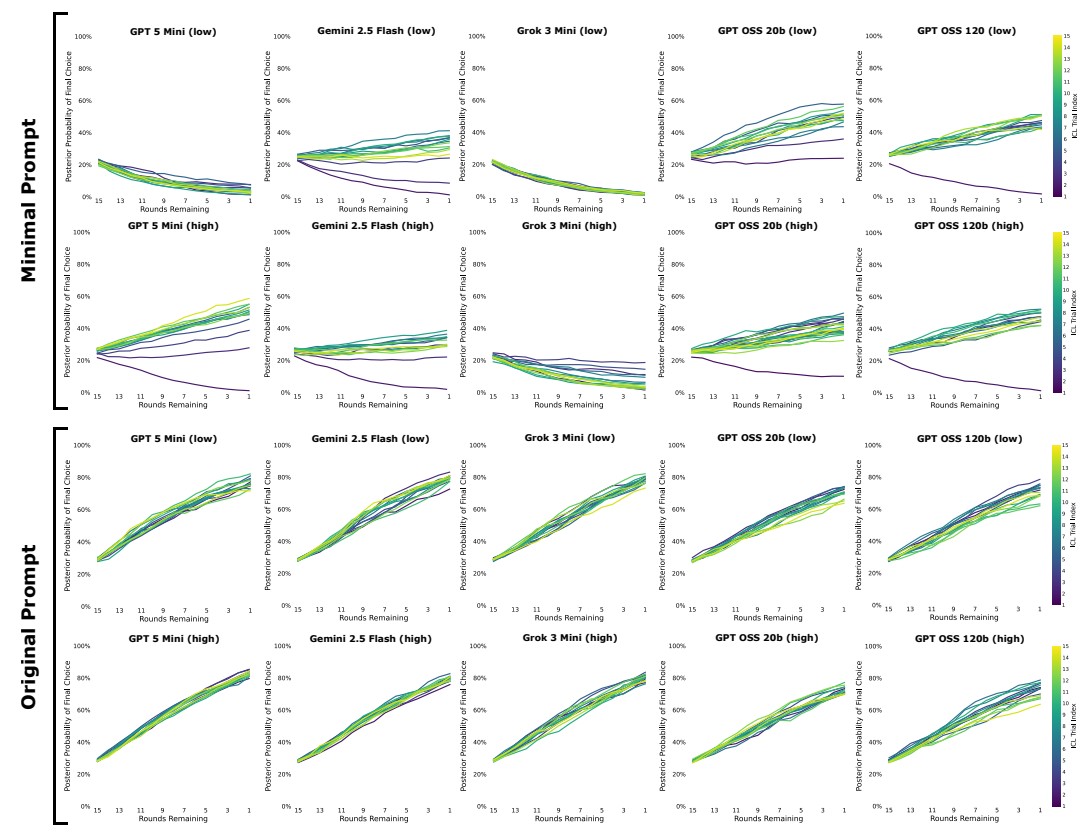

Figure 11: **In-context learning across repeated trials and posterior evolution by reasoning level and prompt.** Models show consistently better evidence integration under the original than the minimal prompt. With high reasoning, the posterior probability of the final choice increases across models and prompts. Additionally, high reasoning reduces across-trial variance in the posterior under the original prompt setting.

# B  OBSERVER MODEL AND GEOMETRIC POLICY UPDATES

This section contains a self-contained derivation of the observer model presented in Section 6 based on a categorical and coupled belief over a set of latent indexes, together with the derivations of the interpolating family of gradients. We consider a single latent discrete variable $z \in \{1, \ldots, K\}$ indicating which arm is currently biased. At the *start of round* $t$, the agent holds a categorical belief $p_t(z)$ over the $K$ possibilities. Since $z$ is discrete, $p_t$ lies on the $(K-1)$-simplex:

$$\sum_{k=1}^{K} p_t(z = k) = 1, \qquad p_t(z = k) \geq 0. \tag{3}$$

When the agent probes arm $i \in \{1, \ldots, K\}$, it observes a Bernoulli outcome $x \in \{0, 1\}$ whose success probability depends on whether the probed arm matches the latent index $z$. We fix the outcome coding so that $x = 1$ denotes *RED* and $x = 0$ denotes *GREEN*. Let $0 < \theta_L < \theta_H < 1$ be the two *likelihood parameters* (emission probabilities): $\theta_H$ is the probability of observing RED when the probed arm is the true latent ($z = i$), and $\theta_L$ is the probability of observing RED when it is not ($z \neq i$). The likelihood is

$$\Pr(x = 1 \mid a = i, z = k) = \begin{cases} \theta_H & \text{if } k = i, \\ \theta_L & \text{if } k \neq i, \end{cases} \tag{4}$$

Here $\theta_H$ and $\theta_L$ are *not* accumulated evidence; they are fixed (or slowly learned) per-trial emission biases. Accumulated evidence appears in the posterior over the latent index, $p_t(z)$, which is updated by Bayes' rule using the likelihood above.

$$\Pr(x = 0 \mid a = i, z = k) = 1 - \Pr(x = 1 \mid a = i, z = k). \tag{5}$$

This coupling through $z$ induces *dependence* across arms: evidence that increases belief in $z = i$ simultaneously decreases belief in $z \neq i$. Alternative belief streams could be used as well to integrate evidence, where the latents would be updated in an independent manner, with the likelihood in this case being given by

$$p\big(x \mid a = i, z_1, \ldots, z_K\big) = z_i^x \big(1 - z_i\big)^{1-x}, \tag{6}$$

i.e., only the probed arm's latent $z_i$ enters the Bernoulli likelihood while the coordinates $\{z_j\}_{j \neq i}$ remain unchanged for that observation. For the *Optimal Observer* fitting to the task in Figure 6, panel **E** and on the behavioral data of the human and LLM games we use the shared rates in Equation 4.

## B.1  PREDICTIVE DISTRIBUTIONS IN THE ACTIVE PROBABILISTIC REASONING TASK

Before acting, the agent can compute, for each candidate arm $i$, the distribution of possible outcomes by marginalizing the likelihood against the current belief. The predictive distribution for arm $i$ is

$$q_{t,i}(x) = \int \Pr(x \mid a = i, z) \, p_t(z) \, dz, = \sum_{k=1}^{K} \Pr(x \mid a = i, z = k) \, p_t(z = k) \tag{7}$$

where the integral equals the sum because $z$ has finite support. Specializing to the Bernoulli case, the success probability has the closed form

$$q_{t,i}(1) = \theta_R + (\theta_R - \theta_G) \, p_t(z = i), \tag{8}$$

and the failure probability is its complement $q_{t,i}(0) = 1 - q_{t,i}(1)$. Suppose the agent hypothetically probed arm $i$ and observed outcome $x \in \{0, 1\}$. The resulting *hypothetical* posterior is obtained by Bayes' rule:

$$p_{t|i,x}(z = k) = \frac{\Pr(x \mid a = i, z = k) \, p_t(z = k)}{\sum_{k'=1}^{K} \Pr(x \mid a = i, z = k') \, p_t(z = k')} \tag{9}$$

After the *actual* action $a_t$ and observation $x_t$, the *real* posterior at the start of round $t+1$ is given by the belief $b_t(z)$

$$b_{t|i}(z = k) = p_{t+1}(z = k) = \frac{\Pr(x_t \mid a_t, z = k) \, p_t(z = k)}{\sum_{k'=1}^{K} \Pr(x_t \mid a_t, z = k') \, p_t(z = k')} \tag{10}$$

A greedy MAP chooser that converts the belief into a deterministic sample uses the posterior mode:

$$\pi_t^{\text{MAP}} = \arg \max_{i \in \{1, \ldots, K\}} p_t(z = i). \tag{11}$$

For the Bayesian agreement, present in Section 3 and Figures 2, 3 we define the agreement score as depending on what an agent which would build this posterior from the evidence would chose in its last round $T$, concretely the score is given by

$$S_T = \frac{1}{N} \sum_{i=1}^{N} \mathbb{I}\left[\mathrm{a}_T = \pi_T^{MAP}\right].$$

(12)

where $a_T$ is the final action taken by the Human or LLM agent.

## B.2 DEFINING A FAMILY OF SCORE-BASED OBSERVERS

At each round, the belief $p_t$ is mapped to a per-arm *score* vector $\phi_t \in \mathbb{R}^K$. We write the Euclidean inner product as $\langle x, y \rangle = \sum_i x_i y_i$ and denote the all-ones vector by $\mathbf{1}$. The generic evidence score defined in panel **E** of Figure 1 for arm $i$ takes the predictive expectation of a utility function $u$ defined on the hypothetical posterior:

$$\phi_{t,i} = \mathbb{E}_{x \sim q_{t,i}}\left[u\left(p_{t|i,x}; i, x\right)\right].$$

(13)

Explicitly, this predictive expectation can be defined as a function of the predictive distribution $q_{t,i}$ as

$$\phi_{t,i} = \int u\left(p_{t|i,x}; i, x\right) q_{t,i}(x)\, dx. = \sum_{x \in \{0,1\}} q_{t,i}(x)\, u\left(p_{t|i,x}; i, x\right)$$

(14)

For the modeling done in Section 6 we chose the following score functions that represent different sampling policies with temperature $\beta > 0$. These scores are inspired from multiple classical behavioural modelling paradigms (Gershman, 2018; Binz et al., 2024) and have been used extensively in computational neuroscience and cognitive psychology to model the behaviour of human subjects under psychophysical tasks. For the MAP agent we define the score function as

$$\phi_{t,i}^{\mathrm{MAP}} = \beta \log p_t(z = i)$$

(15)

which favors the arm with the largest posterior mass and thus exploits current belief, ignoring how informative the next observation might be. Additionally, we define an *Entropy-Minimizing* agent by

$$\phi_{t,i}^{\mathrm{EM}} = -\beta\, H(q_{t,i}).$$

(16)

which prefers arms whose predictive is most peaked (lowest entropy), i.e., choices expected to yield a decisive RED/GREEN outcome regardless of which way it goes.
We define a *Self-Certainty* agent, inspired by (Zhao et al., 2025),

$$\phi_{t,i}^{\mathrm{SC}} = \beta\, D_{\mathrm{KL}}\left(U \| q_{t,i}\right).$$

(17)

which seeks arms whose predictive deviates most from uniform ($1/2$), emphasizing certainty in the immediate observation rather than expected entropy reduction after updating.
Finally, an *Information-Gain* agent is defined by

$$\phi_{t,i}^{\mathrm{IG}} = \beta \sum_{k=1}^{K} p_t(z = k)\, D_{\mathrm{KL}}\left(\Pr(\cdot \mid a = i, z = k) \| q_{t,i}\right).$$

(18)

which prefers arms whose outcomes are expected to most reduce uncertainty about $z$, balancing current belief $p_t(z)$ with the diagnostic gap between $\theta_H$ and $\theta_L$.

## B.3 POLICY PARAMETERIZATION

Actions are sampled from a softmax policy over logits $\theta_t \in \mathbb{R}^K$:

$$\pi_t = \mathrm{softmax}(\theta_t), \qquad (\pi_t)_i = \frac{e^{\theta_{t,i}}}{\sum_{j=1}^{K} e^{\theta_{t,j}}}.$$

(19)

At a given round, with belief (and thus score) held fixed, we consider the linear objective

$$J(\pi) = \langle \pi, \phi_t \rangle,$$

(20)

which will allocate more probability to higher-scored arms, given the information previously sampled. Softmax policies are invariant to adding a constant to all logits, and valid probability updates must preserve normalization. Both constraints are enforced by *centering* the score through an "Advantage"

$$A_t^\phi = \phi_t - \langle \pi_t, \phi_t \rangle \mathbf{1}. \tag{21}$$

The intuition for this definition goes as follows: positive entries imply that the corresponding arm looks better than the one with the current highest probability, whilst negative entries mean "worse than our average," and zeros mean "no change." If the policy already puts weight on the right arms, the average is high and the advantages are small, so updates are mild. If the policy is missing a good arm, that arm gets a large positive advantage and its probability is pushed up quickly. As more evidence arrives and uncertainty drops, score gaps narrow and the whole advantage vector tends toward zero, stabilizing the policy. When predictions are sharper (peaked $q_{t,i}$ or larger temperature $\beta$ in the corresponding score function of an agent), the score gaps widen and advantages grow, producing stronger reallocation; when predictions are ambiguous, scores are flat and the advantages are near zero, so the policy barely moves.

One crucial distinguishing factor of this task, besides the binary sampling outcomes at each arm (or button) is that the cues at each round can become occluded. We represent environmental occlusions with a binary availability mask $m_t \in \{0,1\}^K$, where $m_{t,i} = 1$ if arm $i$ is visible at round $t$ and $m_{t,i} = 0$ otherwise. The policy over *available* arms is the masked softmax

$$\pi_{t,i}^{\mathrm{av}} = \frac{m_{t,i}\, e^{\theta_{t,i}}}{\sum_{j=1}^K m_{t,j}\, e^{\theta_{t,j}}} = \mathrm{softmax}\big(\tilde{\theta}_t\big)_i, \qquad \tilde{\theta}_{t,i} = \begin{cases} \theta_{t,i} & m_{t,i} = 1, \\ -\infty & m_{t,i} = 0. \end{cases} \tag{22}$$

Unavailable arms receive zero probability and no gradient and the advantage is calculated only using the available set. We thus normalize only over visible arms, compare each visible arm's score to the masked average, and update only those logits; hidden arms neither draw probability mass nor receive updates until they reappear.

### B.4 Softmax Jacobian and the Euclidean policy gradient in logit space

The Jacobian of the softmax map $\theta \mapsto \pi$ has entries

$$\frac{\partial \pi_i}{\partial \theta_j} = \pi_i\, (\delta_{ij} - \pi_j), \tag{23}$$

so in matrix form

$$D\pi(\theta) = \mathrm{diag}(\pi) - \pi\pi^\top. \tag{24}$$

Using the chain rule, the Euclidean gradient of $J$ with respect to logits is

$$\nabla_\theta J(\theta_t) = (D\pi)^\top \phi_t = \big(\mathrm{diag}(\pi_t) - \pi_t\pi_t^\top\big)\phi_t. \tag{25}$$

Separating the diagonal and rank-one terms yields

$$\nabla_\theta J(\theta_t) = \mathrm{diag}(\pi_t)\,\phi_t - \pi_t\, \langle \pi_t, \phi_t \rangle = \pi_t \odot \big(\phi_t - \langle \pi_t, \phi_t \rangle \mathbf{1}\big) = \pi_t \odot A_t^\phi. \tag{26}$$

Thus, the Euclidean policy-gradient (PG) step with step size $\alpha > 0$ is

$$\theta_{t+1} = \theta_t + \alpha\,\big(\pi_t \odot A_t^\phi\big). \tag{27}$$

### B.5 Fisher information for softmax and its tangent action

The Fisher information matrix (FIM) Amari and Nagaoka (2000); Mertikopoulos and Sandholm (2016) of a categorical softmax policy equals the softmax Jacobian in this parametrization

$$F(\theta) = \mathbb{E}_{a \sim \pi_\theta}\big[\nabla_\theta \log \pi_\theta(a)\, \nabla_\theta \log \pi_\theta(a)^\top\big] = \mathrm{diag}(\pi) - \pi\pi^\top. \tag{28}$$

We note that the *null direction*, given by

$$F\,\mathbf{1} = 0, \tag{29}$$

implies that shifting all logits together leaves the policy unchanged. Additionally, the *tangent action* for any vector $v$ with $\langle \pi, v \rangle = 0$ (i.e., $v$ lies in the simplex tangent), i.e.

$$F\,v = \mathrm{diag}(\pi)\,v. \tag{30}$$

Since $A_t^\phi$ is tangent by construction, we obtain the identity

$$F A_t^\phi \;=\; \mathrm{diag}(\pi)\, A_t^\phi \;=\; \pi \odot A_t^\phi \;=\; \nabla_\theta J(\theta_t). \tag{31}$$

This work connects with previous work linking replicator flows in statistical game theory which links the underlying evolution of distributions in stochastic environments, such as $k$-armed bandits, under Riemann metric (Harper, 2009; Shahshahani, 1979).

## B.6 NATURAL POLICY GRADIENT AS A RIEMANNIAN GRADIENT

We now connect the Euclidean gradient from the previous subsection to the natural (Riemannian) one and make explicit why the centered score $A_t^\phi$ is the natural ascent direction.
For nearby logits $\theta$ and $\theta + \Delta\theta$, the KL divergence between the corresponding policies has the second–order expansion

$$D_{\mathrm{KL}}\big(\pi_{\theta+\Delta\theta} \,\|\, \pi_\theta\big) \;=\; \tfrac{1}{2}\, \Delta\theta^\top F(\theta)\, \Delta\theta \;+\; o(\|\Delta\theta\|^2), \tag{32}$$

so $F(\theta)$ acts as the local metric (inner product) on logit space.
The natural gradient $\widetilde{\nabla} J$ at $\theta$ is the unique tangent vector that reproduces the directional derivative under this metric, i.e.

$$F \widetilde{\nabla} J \;=\; \nabla_\theta J. \tag{33}$$

From the softmax Jacobian, we already obtained

$$\nabla_\theta J(\theta_t) \;=\; \pi_t \odot A_t^\phi \;=\; \mathrm{diag}(\pi_t)\, A_t^\phi, \tag{34}$$

where $A_t^\phi = \phi_t - \langle \pi_t, \phi_t\rangle \mathbf{1}$ satisfies $\langle \pi_t, A_t^\phi\rangle = 0$ (so $A_t^\phi$ is tangent).
From the Fisher's tangent action (shown earlier), any tangent $v$ obeys $Fv = \mathrm{diag}(\pi)\, v$. Applying this to $A_t^\phi$ gives

$$F A_t^\phi \;=\; \mathrm{diag}(\pi_t)\, A_t^\phi \;=\; \nabla_\theta J(\theta_t), \tag{35}$$

where the last equality used (34).
Comparing (33) and (35) shows that $A_t^\phi$ solves the defining equation of the natural gradient. Because $A_t^\phi$ is tangent (orthogonal to the null direction of $F$), it is the canonical solution:

$$\widetilde{\nabla} J \;=\; A_t^\phi. \tag{36}$$

A natural-gradient (NG) step of size $\alpha > 0$ updates the logits by

$$\theta_{t+1} \;=\; \theta_t \;+\; \alpha\, A_t^\phi \;=\; \theta_t \;+\; \alpha\big(\phi_t - \langle \pi_t, \phi_t\rangle\, \mathbf{1}\big). \tag{37}$$

This step is (i) baseline-invariant (it ignores the shift direction), (ii) normalization-preserving in policy space (because it is tangent), and (iii) curvature-aware: it "undoes" the $\mathrm{diag}(\pi_t)$ scaling present in the Euclidean gradient, yielding motion matched to the local KL geometry.

## B.7 AN INTERPOLATING FAMILY OF LOGIT GEOMETRIES

To compare traversal geometries, endow the logit space with the diagonal metric

$$G_\gamma(\theta) \;=\; \mathrm{diag}\big(\pi^{1-\gamma}\big), \qquad \gamma \in [0, 1]. \tag{38}$$

At $\gamma = 1$ this is the Euclidean metric on logits ($G_1 = I$). At $\gamma = 0$ it is $G_0 = \mathrm{diag}(\pi)$, which coincides with the Fisher action on the simplex tangent.
The steepest-ascent direction under $G_\gamma$ solves the trust-region problem

$$\max_u \; \nabla_\theta J^\top u \quad \text{subject to} \quad u^\top G_\gamma u \;\leq\; \varepsilon. \tag{39}$$

By Cauchy–Schwarz in the $G_\gamma$ inner product, the optimizer aligns with $G_\gamma^{-1} \nabla_\theta J$. Using $\nabla_\theta J = \pi \odot A_t^\phi$ and the diagonal form of $G_\gamma^{-1}$,

$$G_\gamma^{-1} \nabla_\theta J \;=\; \mathrm{diag}\big(\pi^{\gamma-1}\big)\, (\pi \odot A_t^\phi) \;=\; \pi^\gamma \odot A_t^\phi. \tag{40}$$

Thus the *interpolated* logit update with step size $\alpha$ is

$$\theta_{t+1} \;=\; \theta_t \;+\; \alpha\, \big(\pi_t^\gamma \odot A_t^\phi\big), \qquad \gamma \in [0, 1]. \tag{41}$$

The factor $\pi_i^\gamma$ acts as a state-dependent throttle: larger $\gamma$ damps motion on low-probability coordinates; smaller $\gamma$ removes that damping.

Under this definition we can then fully recover the classical gradients. When $\gamma = 1$, we have $G_1 = I$ and $G_1^{-1} \nabla_\theta J = \nabla_\theta J = \pi \odot A_t^\phi$, so the update is exactly the Euclidean policy-gradient step:

$$\theta_{t+1} \;=\; \theta_t \;+\; \alpha \, (\pi_t \odot A_t^\phi). \tag{42}$$

When $\gamma = 0$, we have $G_0 = \mathrm{diag}(\pi)$ and $G_0^{-1} \nabla_\theta J = \mathrm{diag}(\pi)^{-1}(\pi \odot A_t^\phi) = A_t^\phi$, so we recover the natural policy-gradient step:

$$\theta_{t+1} \;=\; \theta_t \;+\; \alpha \, A_t^\phi. \tag{43}$$

Hence the interpolation continuously connects Euclidean PG and NPG in logit space.

## C  MODEL FITTING DETAILS

**Hyperparameters are explored on grids.** We fit observer models to 15-round choice sequences from LLMs and humans. Each game yields $(m_{t,i}, a_t, r_t)$ for $t = 1..15$ with $m_{t,i}$ the available cue mask defined in Equation 22, $a_t \in m_{t,i}$ the chosen cue, and $r_t \in \{0, 1\}$ (RED = 1). Each cue is processed with the posterior defined in Equation 4. After each round we update the corresponding $\phi_t$ for a given observer of the set $\{MAP, EM, SC, IG\}$, set $A_t = \phi_t - (\pi_t^\top \phi_t)\mathbf{1}$, and update $\theta_{t+1} = \theta_t + \alpha \, (\pi_t^\gamma \odot A_t)$. For likelihood seeding, we set a cue-wise $\tilde{\mu}$ with the biased cue at 0.9 and others at 0.5, which is the environment shared in the human version of the task.

**Hyperparameters are explored on grids**. We use $\gamma \in \{-3, 3\}$, $\alpha \in \{0.01, 0.5\}$, and $\beta \in \{-15, 15\}$. For each observer and each parameter triple $(\gamma, \alpha, \beta)$, we iterate over 100 games sampled over a set of 10 random seeds for an agent group (LLM model or human cohort), run the 15-round update, compute per-game log-likelihood, and average across games. The winning parameters are chosen over this set by the average log-likelihood over the grid per agent, and observer, and seed are reported for their respective observer models in Figure 6 **D**.

**Fitting results**. The pseudo-$R^2$ was used to quantify the explanatory power of each fitted and refitted model (Figures 6 **B,C**) relative to a random baseline. For a given synthetic dataset and candidate observer, the refitting procedure yields the best log-likelihood $LL_{\mathrm{model}}$ after grid search over $(\gamma, \alpha, \beta)$. As a baseline, we compute $LL_{\mathrm{random}}$, the log-likelihood of a uniform policy that assigns equal probability to each currently available arm at every round. We report the pseudo-$R^2$ given by $R^2 = 1 - \frac{LL_{\mathrm{model}}}{LL_{\mathrm{random}}}$. To account for stochasticity from both environment sampling and parameter initialization, the $R^2$ calculation was repeated across multiple seeds. Specifically, for each (agent, generating model), we ran 10 simulations of $n_{\mathrm{games}} = 200$ games each, refitted candidate observers. We show in Figure 6 **C** the correspoding winning seed $R^2$ matrix.

## D  REINFORCEMENT LEARNING AGENT DETAILS

We train a reinforcement learning (RL) to perform the task. We use Proximal Policy Optimization (PPO)Schulman et al. (2017) to train the agent. Observations are a 13-dimensional vector comprising cue–color counts (8 total, 4 choices for two possible colors per button), a phase indicator (sampling vs. decision), and a binary availability mask over cues (4). Environment parameters (cue bias, hidden cues, trial length, etc.) follow that of the task with trial length selected uniformly at random from possible game lengths. We assign rewards of $+100$ for a correct final choice, 0 for valid sampling moves, and $-1$ for selecting an unavailable cue. Invalid selections advance the round without informative evidence, incentivizing correct cue selection along with final selection.

We performed a grid search for optimization and architecture parameters that maximizes correct cue selection. The policy is an MLP with ten hidden layers of 64 units and *tanh* activations. Training uses PPO for $5 \times 10^6$ environment steps with learning rate $2 \times 10^{-5}$, minibatch size 128, each iteration collects $T = 2048$ on-policy steps, and 10 optimization epochs per iteration. We set $\gamma = 0.99$, $\lambda_{\mathrm{GAE}} = 0.95$, and clipping $\varepsilon = 0.2$, with entropy and value losses weighted by 0.01 and 0.5 respectively.

## E  PROMPTING STRUCTURE USED FOR THE LLMs IN THE TASK

Here we outline the two main prompts used to evaluate the language models. The `original prompt` includes task, mechanics, and scoring. States bias structure (one cue 90/10, others 50/50). Mentions cue availability dynamics (cues may disappear; at least one always active). Enforces

strict response format per round (single letter, no punctuation). Final query targets the cue with the highest RED ratio. Provides feedback taxonomy (correct/incorrect/invalid) with reveal of the biased cue. Specifies payoff: +100 for correct, 100 for wrong. This is the richest instruction set for the task, with explicit probabilistic environment and payoff structure. Stronger priming toward bias identification rather than generic correctness. Final feedback frames success as detecting the "biased cue," not just a "correct cue.". The models are evaluated in an interactive environment and fed round and trial related information based on their choices across whole trials or, in the case of the ICL experiments in Section 5, multiple trials.

---

**Original prompt**

```
Task
- You will play a game with {n_rounds} rounds.
- In each round, some cues are available: A, B, C, D.
- One cue is biased: 90% one color / 10% the other. The others are
    50/50.
- Available cues may disappear at random, but at least one is always
    active.
- Each round, respond with exactly one letter (A, B, C, or D). No
    markup or punctuation.
- After {n_rounds} rounds, identify the biased cue.
- Scoring: Correct +100 points, Wrong 100 points.

Round prompt
- Trial {current_trial}, Round {current_round}: Available cues {
    available_cues}. Which do you choose? Respond with exactly one
    letter: A, B, C, or D.

Round feedback
- Trial {current_trial}, Round {current_round}: Available cues {
    available_cues}. You chose {current_answer} and saw {result_text}.

Final decision prompt
- Trial {current_trial}: Based on all observed colors, which cue {
    letters} had the highest ratio of RED? Respond with exactly one
    letter: A, B, C, or D.

Final feedback
- Trial {current_trial}: Based on all observed colors, which cue {
    letters} had the highest ratio of RED?
  You chose {current_answer} which was {feedback}.
  You received {score} points.

Feedback labels
- Correct: the biased cue
- Incorrect: not the biased cue. The biased cue was {biased_quadrant}
- Invalid: an invalid choice
```

---

The following prompt is the `minimal prompt` used for the ICL Section 5. Covers task and per-round response rule only. Lists cues (A–D). Omits bias mechanics, cue disappearance dynamics, and scoring. Final query asks for a choice without RED-ratio framing. Includes feedback taxonomy (correct/incorrect/invalid) with reveal of the "correct cue." No payoff details. No probabilistic structure or availability dynamics are provided. Neutral objective framing ("choose") rather than bias detection. Simplified feedback semantics ("correct cue" vs "biased cue").

---

**Minimal prompt**

```
Task
- You will play a game with {n_rounds} rounds.
```

```
- In each round youll see cues: A, B, C, D.
- Each round, respond with exactly one letter (A, B, C, or D). No
    markup or punctuation.
- After {n_rounds} rounds, make a final decision.

Round prompt
- Trial {current_trial}, Round {current_round}: Available cues {
    available_cues}. Which do you choose? Respond with exactly one
    letter: A, B, C, or D.

Round feedback
- Trial {current_trial}, Round {current_round}: Available cues {
    available_cues}. You chose {current_answer} and saw {result_text}.

Final decision prompt
- Trial {current_trial}: Based on all observed colors, which cue {
    letters} do you choose?
    Respond with exactly one letter: A, B, C, or D. I choose:

Final feedback
- Trial {current_trial}: Based on all observed colors, which cue {
    letters} do you choose?
    You chose {current_answer} which was {feedback}.
    You received {score} points.

Feedback labels
- Correct: the correct cue
- Incorrect: not the correct cue. The correct cue was {biased_quadrant
    }
- Invalid: an invalid choice
```

## F    TESTING A SUBSET OF MODELS ON AN INSTRUCTED BANDIT TASK

In order to further validate our methodology we tested a subset of models on a simpler task. One of the defining features of the the active probabilistic reasoning task presented in Section 3 is that it requires two explicit levels of information and decision making: *sampling* and *inference*. In order to confirm that the structure of this novel task is able to accurately grade performance outside of the distribution the available datasets, such as *Psych101*Binz et al. (2024) which contain over 100k trials on a multitude of psychophysics like tasks, we decided to test our pipeline on an **Instructed Bandit Task** Su et al. (2025) with rewards given at each arm sampled from a gaussian $\mathcal{N}(x; \mu, 10)$ and means vector $\mu = [10, 20, 40, 60, 80]$. We chose *Centaur70B-Instruct* model Binz et al. (2025), a model that was explicitly fine-tuned on *Psych-101* and for which the performance on our benchmark was slightly above random chance, and compare its performance with a smaller *Qwen-14B* model and the more recent reasoning architecture of the *GPT-OSS-20B*, on *LOW* and *MEDIUM* level of reasoning. We ran this experiment with 100 trials per model and each trial consisting of 20 rounds.

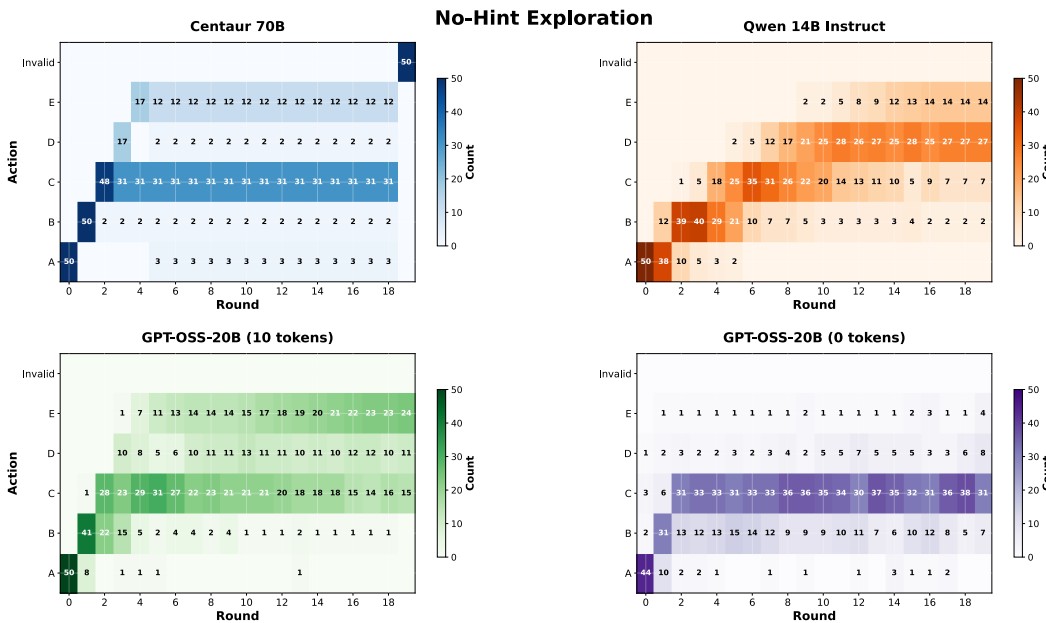

Figure 12: **Performance of Centaur70B vs other models on the instructed bandit task**: Models struggle to chose the highest reward option **E** when there's no hint for exploration.

Surprisingly, with the original prompt used in Su et al. (2025), which also tested Centaur against a Bayesian agent, the model collapses at arm "C", not being able to chose the correct arm most of the time, while *GPT-OSS-20B* shows a more active exploration profile.

**Instructed bandit prompt**

```
– Task
You will play a game with {n_rounds} rounds.
Five arms are available each round: A, B, C, D, E.
Each arm has a fixed but unknown expected reward in [0,100].
Each pull returns a stochastic reward around that arms expected value.
All arms are always available. Values reset each new game.
Goal: maximize total reward points across {n_rounds}.

– Round prompt

Round {current_round}/{n_rounds}: Choose an arm to sample.
Available arms: A, B, C, D, E
```

```
Which arm do you choose? Respond with exactly one letter: A, B, C, D,
    E.

- Round feedback

Round {current_round}/{n_rounds}: You chose arm {current_answer} and
    received {reward:.2f} points.
Running total: {total_reward:.2f} points.

- Final decision prompt

You have completed all {n_rounds} rounds and earned {total_reward:.2f}
    total points.
Based on your experience, which arm has the highest expected reward?
Available arms: A, B, C, D, E
Your answer (one letter only):

- Final feedback

Your total reward: {total_reward:.2f} points

- Feedback labels
Correct: Correct! You successfully identified the optimal arm.
Incorrect: Incorrect. The arm with the highest expected reward was {
    optimal_arm}.
Invalid: not specified
```

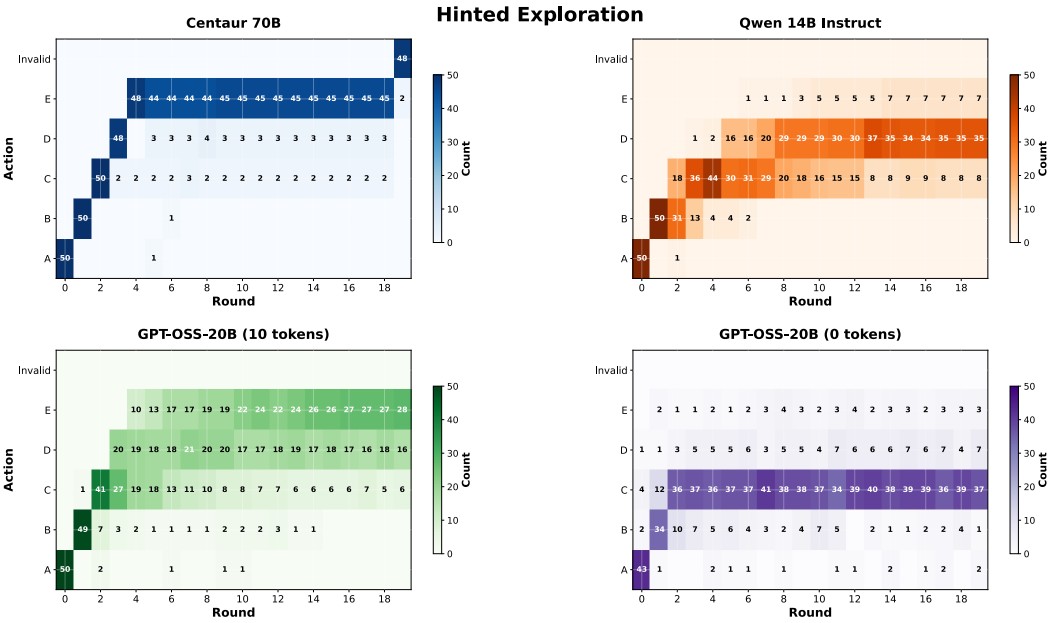

Figure 13: **Performance of Centaur70B vs other models on the instructed bandit task**: Now with the hint to explore all arms as much as possible, both Centaur-70B and GPT-OSS-20B with *LOW* reasoning are able to find the right arm. *Qwen-14B* and *GPT-OSS-20B* without reasoning are not able to pick this option with the same frequency.

We further tested by adding a more explicit hint regarding exploration, which showed a visible increase of performance. By adding the following **hint:** *Explore all arms as much as possible in order to find the highest reward*, both *Centaur* and *GPT-OSS-20B* were now able to reach more frequently to the right arm $E$, which implies some prompting related modulation of the underlying

active sampling policy of these models. Another distinguishing factor is the fact that this task contains a *running score* of the points gathered across a single trial of 20 rounds, which might allow models to more easily steer their choices. However, even in variants of our task where we had kept a running score we didn't observe an improvement in performance and as such did not use this task variant in the main version of the task present in the paper.

