# OpenReview forum: "Active probabilistic reasoning in humans and language models"
_ICLR.cc/2026/Conference — ICLR 2026 Conference Withdrawn Submission_

### Official Review · Reviewer_u38v · 2025-10-16

**Soundness:** 2
**Presentation:** 2
**Contribution:** 1
**Rating:** 2
**Confidence:** 4

**Summary:**

This paper examines LLM performance using a simplified four-armed bandit task. In the task, one arm produces binary outcomes with a 90/10 bias, while the remaining three arms yield unbiased 50/50 outcomes. The authors tested both LLMs and human participants on this task and found that some LLMs achieved performance levels comparable to humans. A gap relative to Bayes-optimal agents appears visually evident, although the paper does not present an explicit statistical test of this performance difference. The analysis focuses primarily on correlations between observed behaviors and those of Bayes-optimal agents. Based on the behavioral data from both LLM and human participants, the authors fit a cognitive model and report that human updating behavior is better characterized by maximum-a-posteriori sampling, whereas LLM behavior aligns more closely with minimizing posterior entropy across options.

**Strengths:**

The paper is generally easy to follow, and the bandit task is clearly described and straightforward to understand. The visualizations are particularly well executed, with figures that are both clear and effective.

**Weaknesses:**

Main comments:
- **Lack of novelty:** The bandit task is interesting, but as the authors themselves acknowledge, Su, Ho, & Gurericks (2025) have already tested similar bandit tasks on LLMs and fitted cognitive models to capture their behavior. More broadly, bandit tasks are extensively studied in LLMs. As a result, the present work does not appear to provide substantially new insights, particularly from the simplified four-armed bandit setting.
- **Mechanistic modeling of LLM behavior:** One possible avenue for improvement would be to develop a more mechanistic account of how LLMs produce behavior in this task. While fitting cognitive models is valuable, it rests on assumptions that may not align with the underlying generative process of LLMs. Mechanistically, LLMs generate the next token conditional on all prior tokens. Linking this conditional probabilistic distribution to the observed bandit behavior could yield more novel insights. More fundamentally, LLM agents lack the form of agency that human participants bring to the task. For instance, humans can be incentivized differently, and we can expect systematic changes in their behavior. LLMs, by contrast, may not respond to incentives in the same way. Nonetheless, their behavior could be better understood by leveraging the fact that their generated outputs are always conditional on the prefix context.

Minor comments:
- Line 425: "bayesian" --> "Bayesian"
- Line 1463: citations in Appendix F are not formatted properly

**Questions:**

NA

---

### Official Review · Reviewer_eBEx · 2025-10-30

**Soundness:** 3
**Presentation:** 3
**Contribution:** 3
**Rating:** 6
**Confidence:** 2

**Summary:**

Introduces an active probabilistic reasoning benchmark separating sampling from inference via an occluded multi-armed task. Across >55k LLM and 5k human trials, top models match/exceed humans in Bayesian inference but lag in sampling; more reasoning aids inference, not sampling. Observer-model fits: humans ~MAP; LLMs entropy-driven.

**Strengths:**

1. Four-armed occluded task cleanly separates sampling vs. inference; diagnostic process signals; simple, reproducible, and extensible design.

2. Large-scale, unified human–LLM comparison with three complementary metrics and two manipulations (reasoning effort; ICL original/minimal); clear findings: inference ≈ MAP, sampling weaker; some degradation under original, gains with minimal + high reasoning.

3. Observer-model fitting unifies MAP/entropy/information-gain strategies; shows humans ≈ MAP, top LLMs ≈ entropy-driven; provides an actionable framework for improving active sampling.

**Weaknesses:**

The paper offers profound insights under a single occlusion condition. Further systematic comparisons across different occlusion conditions (e.g., no occlusion, weak vs. strong occlusion) could potentially reveal additional dimensions of the strategic differences.

**Questions:**

The paper finds that humans tend to use MAP (maximum a posteriori) sampling, whereas top LLMs favor entropy-driven sampling. What potential reasons might underlie this difference?

---

### Official Review · Reviewer_zH7D · 2025-10-30

**Soundness:** 3
**Presentation:** 1
**Contribution:** 2
**Rating:** 2
**Confidence:** 3

**Summary:**

This paper is primarily concerned with comparing humans' and LLMs' abilities to sample effectively and evaluate the evidence they have collected (inference), with the stated goal of evaluating LLM mastery of sequential reasoning tasks. To facilitate the comparison of human and LLM abilities, the paper introduces a new task in which the goal is to determine which of several buttons is biased: participants collect evidence for several rounds and then make a final judgment in a single inference round. The data collected for several LLMs on this task is then compared with human data from a study conducted by the authors, with the conclusion that while LLMs are equally as good, if not better, at inference, humans employ better and different sampling strategies.

**Strengths:**

The paper’s focus on isolating sampling from inference in evaluating LLM decision is original, distinguishing it from other LLM probabilistic reasoning experiments. The finding that the difference in human and LLM performance on sequential reasoning tasks seems to stem from differences in sampling strategies rather than issues with inference seems like it could be a point of significance if the discussion of the findings in the paper were better improved upon. In terms of its explanation of methods and results, the paper was generally clear and easy to follow.

**Weaknesses:**

Overall, while the paper generally contains sufficient explanation of the methods employed and the subsequent results, the motivation for these methods and the implications of the results are not well discussed. A few examples are:

1. Although the details of the experiments performed in Section 5 were clear, their relationship with the stated goals of the paper was more tenuous. This section moves away from comparing model behavior with human data to a discussion of the impact of model reasoning effort on ICL ability. The connection with the paper’s overall vision is not as clear as it could be.
2. Section 6 compares LLM and human behavior with a number of observer models, but the fit of even the best agent/observer pairing is not particularly strong, making these results not especially convincing. Even if one accepts that humans are more MAP-like in their strategy while LLMs tend more towards EM or SC, there is no discussion of why this discovery is important.

Additionally, the current survey of related work is quite limited and in many cases explicit connections are not made to the main text. It would be helpful to make occasional connections between the topics listed in the related work and the main goals and content of the paper.

**Questions:**

1. Please clarify the motivation behind Section 5 and its connections with the rest of the paper.
2. What do the different sampling strategies discovered in Section 6 reveal about human and LLM models of cognition?
3. The conclusion that LLMs display human-like evidence integration seems to stem from the observation that their posterior dynamics are qualitatively similar. Is there any other way to quantify or define ‘human-like’ evidence integration?
4. How does this work compare to other sequential reasoning experiments in LLMs? Other than the isolation of sampling/inference, what other features set this work apart? Improving the discussion of related work could be accomplished by either adding more detail about the references already listed in the paper or by connecting this work to a broader range of related topics.

---

### Official Review · Reviewer_WBQ6 · 2025-11-01

**Soundness:** 3
**Presentation:** 2
**Contribution:** 3
**Rating:** 4
**Confidence:** 4

**Summary:**

This paper investigates whether large language models (LLMs) can match human cognitive capabilities in sequential probabilistic reasoning. The authors design a novel active reasoning task that decouples two key components of decision-making: sampling (evidence gathering) and inference (evidence evaluation). Testing a diverse set of LLMs, they find that while several frontier models achieve human-level performance overall, no model surpasses skilled human players. Through a Bayesian modeling framework, the authors reveal a dissociation in strategies: humans tend toward maximum-a-posteriori (MAP) sampling (selecting the option most likely to be the best choice among four alternatives), whereas top-performing LLMs tend to minimize posterior entropy, choosing options expected to yield the most certain outcomes. The authors also examine whether LLMs can improve through in-context learning, finding that only a subset of high-performing models can learn from choice outcomes alone.

**Strengths:**

1. The paper introduces fine-grained behavioral measures to reveal the dynamics of sequential decision-making in both humans and LLMs. For instance, the authors track the evolving posterior probability of the final choice to estimate sampling strategies, providing a level of analytical detail rarely seen in LLM-human comparisons.
2. The systematic reporting of how different levels of reasoning influence sampling behavior offers insights into the computational mechanisms underlying model performance.
3. The investigation of meta-learning abilities—specifically, whether LLMs can extract and use prior environmental information through in-context learning to improve future performance—represents a novel contribution to understanding LLM capabilities beyond single-task performance.

**Weaknesses:**

1. **Limited characterization of the inference phase.** While the authors claim to have designed a task separating sampling and inference, the paper focuses almost exclusively on sampling, leaving the inference component underexplored. The task design appears to minimize the inference demands, making this effectively a sampling-focused study rather than a balanced investigation of both components. Notably, Figure 3B shows superior human performance in corrected success rates while uncorrected performance remains comparable to LLMs, suggesting humans may actually be worse at inference (evidence summarization) than LLMs. A formal model of the inference phase would clarify this apparent paradox and strengthen the paper's central claim about component separation.
2. **Lack of interpretability regarding reasoning benefits.** The paper documents that reasoning improves LLM performance but provides limited insight into why or how this occurs. Without mechanistic explanation, it remains unclear whether reasoning helps models implement better sampling policies, improve inference accuracy, or both, or just reduce random errors, and through what computational processes these improvements emerge.
3. **Insufficient model validation and baselines.** The paper does not demonstrate whether the best-fitting models can actually replicate human or LLM behavior, raising concerns about model adequacy. A model may appear superior simply by being "the best among poor alternatives" without accurately capturing the underlying processes. More critically, the absence of optimal and random performance baselines makes it impossible to assess the absolute quality of human and LLM performance. Related to this, MAP sampling (greedily choosing the expected best option) appears suboptimal for this task—a better strategy would maximize the probability of discovering the best option through sampling. If humans use MAP and still outperform LLMs, this suggests either that LLMs adopt even worse strategies or that the task structure particularly favors MAP. Without these reference points, the significance of the human-LLM performance gap remains ambiguous.
4. **Limited task diversity and generalizability.** The experimental design uses a specific environmental structure (one good option, three equally poor options) that may particularly favor MAP sampling strategies. This raises concerns about whether the conclusions generalize beyond this specific setup. Testing the human advantage across varied task structures (e.g., different reward distributions, varying numbers of options) would be necessary to establish whether LLMs genuinely lag in sampling or whether humans simply excel in this particular configuration. The narrow task design limits confidence in the broader claims about LLM cognitive capabilities.
5. **Incomplete analysis of in-context learning.** While the paper reports which models can learn from minimal prompts, it does not propose or test formal models that could capture the learning mechanisms underlying successful in-context learning. Understanding what computational processes enable certain models to extract and apply task structure from experience would significantly strengthen this contribution.

**Questions:**

### Q1: Inference phase characterization and modeling
Can you formally model the inference phase to test how humans and LLMs differ in evidence aggregation? This would validate your component separation claim and potentially reveal complementary strengths between humans and LLMs.
### Q2: Performance baselines and model validation
Can you compute the optimal sampling policy (e.g., via dynamic programming or RL) and establish random baselines? Additionally, MAP sampling appears suboptimal—optimal strategies should maximize the probability of discovering the best option, not greedily select the current best. Why do humans outperform LLMs despite using this seemingly suboptimal strategy? Please also provide model recovery analyses or posterior predictive checks demonstrating that your best-fitting models actually replicate observed human and LLM behavioral patterns.
### Q3: Generalizability across task structures
Your task structure (one good option, three equally poor options) may particularly favor MAP sampling. Can you test whether findings generalize to other configurations, such as different reward distributions (e.g., two good/two poor, graded rewards) and varying numbers of options? If the human advantage diminishes or reverses under alternative configurations, this would suggest task-specific rather than general cognitive differences.
### Q4: Mechanistic understanding of reasoning and learning
Can you analyze whether reasoning improves sampling policy selection, inference accuracy, or both? For instance, do reasoning-enabled models converge toward different sampling strategies or show better evidence integration? Similarly, for in-context learning, can you propose and test formal models of the learning mechanisms? What enables some models to update priors or adjust policies while others cannot? Comparing successful versus unsuccessful models could reveal critical architectural or training features.

### Additional Minor Comments
* The writing would benefit from further polishing, such as improving the logical flow of the abstract, the equation notations, and figure presentation.
* The term "sequential reasoning" may be a little confusing. At first sight, I thought it suggests multi-step reasoning rather than sequential decision-making.
* Several technical issues require attention: In Equation 1, the index *i* in the definition of S_T should likely be *t*, and a_*T* should likely be a_*t*. Additionally, two equations are both labeled as Eq. 1 and should be numbered separately.
* Figure presentation needs standardization: panel labels should consistently appear in the top-left corner rather than mixing left and right positions, as the current inconsistency creates confusion. In Figure 5D & 5G, the color scheme differs from other panels, and the relationship between the x-axis and color scale is unclear—both appear to represent ICL trial index, which seems impossible.
* Figure 6C would be more informative showing the proportion of data best fit by the true generative model rather than mean R², because the current presentation does not reveal whether substantial proportions of models might be misidentified.

---

### Note · Authors · 2025-12-03

**Comment:**

We thank the reviewers for their careful reading of our work and for the many constructive comments. Across the reviews, a central and valid concern is that, while the empirical patterns are robust, our current manuscript does not yet offer a sufficiently developed mechanistic model that jointly explains sampling and inference in humans and LLMs. We are actively working on such a model, but these results are still preliminary. To avoid presenting incomplete analyses, we have decided to withdraw the submission and to substantially rework the paper before resubmitting it elsewhere. Below we briefly summarize how we plan to address the main points raised by each reviewer; these remarks are meant as an acknowledgement of the helpful feedback rather than as a full rebuttal with new results.

**Reviewer WBQ6.**
We agree that our analysis of the inference abilities of humans and LLMs was under-developed in the current version. In the revised line of work, we are building a mechanistic model that jointly captures the processes of sampling and inference. We validate these models with psychophysical kernels and logistic regression approaches, in particular to quantify how humans and LLMs weight evidence at each step and then act on it. We also want to clarify here that the requested random and optimal baselines (including a dynamic-programming / RL agent close to the Bayes-optimal ceiling) were already included in the current version of the manuscript.

**Reviewer zH7D**.
We acknowledge that the motivation and placement of the section on reasoning/ICL section were not clearly tied to the main question of disentangling the contributions of sampling and inference to behavior. In the reworked manuscript, we will exploit our newly developed mechanistic model of sampling and inference to characterize in more detail what aspects of the computations underlying the observed behaviors are affected by reasoning and can be improved through ICL.

**Reviewer eBEx.**
We agree that occlusion is a central design choice and that our current empirical exploration is limited to a single occlusion regime. The next version of the work will more clearly justify this regime as a deliberate way to separate sampling from inference. Your question about why humans are more MAP-like while LLMs are more entropy-driven has directly informed our ongoing mechanistic analyses.

**Reviewer u38v.**
We appreciate the concern about novelty relative to existing LLM bandit work such as Su, Ho, & Gureckis and related papers. Our focus is not on introducing yet another bandit benchmark, but to (1) exploit our novel task (which differs in key aspects from regular multi-armed bandit tasks) to disentangle the contributions of sampling and inference to behavior; and (2) compare humans and LLMs within a single modeling framework. Following the reviewer’s suggestions, we are also exploring more closely the link between cognitive-level models and the LLM’s conditional next-token distribution.
We are grateful to all reviewers for helping us clarify the limitations of the current manuscript and for shaping the direction of the next, more mechanistically grounded version of this work.

**Withdrawal Confirmation:**

I have read and agree with the venue's withdrawal policy on behalf of myself and my co-authors.